# Structural insights into ubiquitin chain cleavage by *Legionella* ovarian tumor deubiquitinases

Sangwoo Kang[1], Gyuhee Kim[2], Minhyeong Choi[1], Minwoo Jeong[1], Gerbrand J van der Heden van Noort[3] ⬚,
Soung-Hun Roh[2], Donghyuk Shin[1] ⬚

Although ubiquitin is found only in eukaryotes, several pathogenic bacteria and viruses possess proteins that hinder the host ubiquitin system. *Legionella*, a gram-negative intracellular bacterium, possesses an ovarian tumor (OTU) family of deubiquitinases (Lot DUBs). Herein, we describe the molecular characteristics of Lot DUBs. We elucidated the structure of the LotA OTU1 domain and revealed that entire Lot DUBs possess a characteristic extended helical lobe that is not found in other OTU-DUBs. The structural topology of an extended helical lobe is the same throughout the Lot family, and it provides an S1′ ubiquitin-binding site. Moreover, the catalytic triads of Lot DUBs resemble those of the A20-type OTU-DUBs. Furthermore, we revealed a unique mechanism by which LotA OTU domains cooperate together to distinguish the length of the chain and preferentially cleave longer K48-linked polyubiquitin chains. The LotA OTU1 domain itself cleaves K6-linked ubiquitin chains, whereas it is also essential for assisting the cleavage of longer K48-linked polyubiquitin chains by the OTU2 domain. Thus, this study provides novel insights into the structure and mechanism of action of Lot DUBs.

## Introduction

Protein ubiquitination, a post-translational modification, regulates various cellular events in eukaryotes (Yau & Rape, 2016); thus, the disruption of the ubiquitination system leads to severe problems in humans, including cancer, Parkinson's disease, and neurodegeneration (McNaught et al, 2001; Swatek & Komander, 2016; Schmidt et al, 2021; Tomaskovic et al, 2022). The canonical addition of ubiquitin is tightly regulated by the activating (E1), conjugating (E2), and ligating enzymes (E3) (Komander & Rape, 2012; Popovic et al, 2014; Swatek & Komander, 2016; Yau & Rape, 2016; Dikic & Schulman, 2022; Mukherjee & Dikic, 2022). Proteins can be ubiquitinated by monoubiquitin or polyubiquitin chains, and the linkages are formed via the seven lysine residues or the N-terminal methionine residue of ubiquitin (Komander & Rape, 2012; Mattiroli & Sixma, 2014; Swatek & Komander, 2016; Tomaskovic et al, 2022). Different linkage types of ubiquitin polymers are involved in different cellular events, such as mitophagy and DNA damage repair (K6- and K27-linked chains), proteasomal degradation (K11- and K48-linked chains), innate immunity (K27-, K33-, and K63-linked chains), cell cycle regulation (K29-linked chains), protein trafficking (K33- and K63-linked chains), and the NF-kB signaling pathway (M1-linked chains) (Swatek & Komander, 2016).

Thus, tight regulation of the proper level of ubiquitination is crucial for cellular viability. ~100 deubiquitinases (DUBs) have been reported in humans; they cleave the isopeptide bond between ubiquitin and the target substrates and counteract the E3 ligases (Komander et al, 2009; Clague et al, 2019). Based on their structure and underlying mechanisms, DUBs are classified into seven different families, namely, ubiquitin-specific protease, ovarian tumor (OTU) domain, Jab1/Mov34/Mpr1Pad1 N-terminal domain (JAMM [MPN]), Machado–Joseph domain (MJD [Josephin]), ubiquitin C-terminal hydrolases, zinc finger with UFM1-specific peptidase domain protein (ZUSFP/ZUP1), and motifs interacting with the novel ubiquitin-containing novel DUB family (MINDY) (Abdul Rehman et al, 2016; Haahr et al, 2018; Hermanns et al, 2018; Hewings et al, 2018; Kwasna et al, 2018; Clague et al, 2019). Except for zinc-containing metalloprotease JAMM (MPN), other DUBs are cysteine proteases (ubiquitin-specific protease, OTU, MJD [Josephin], ubiquitin C-terminal hydrolases, ZUSFP, and MINDY). Among these, the OTU family DUBs exhibit ubiquitin linkage specificity (Mevissen et al, 2013; Mevissen et al, 2016; Mevissen & Komander, 2017). For example, OTUD1 and OTULIN specifically cleave K63-linked (Mevissen et al, 2013) and M1-linked (Keusekotten et al, 2013) chains, respectively. OTU family DUBs have a structurally similar OTU domain for catalysis, but additional S1 and S2 binding sites, ubiquitin-binding domains, and sequence variations at the His loop and the variable (V) loop affect their linkage specificity (Mevissen et al, 2013).

[1]Department of Systems Biology, College of Life Science and Biotechnology, Yonsei University, Seoul, Republic of Korea   [2]School of Biological Science, Institute of Molecular Biology and Genetics, Seoul National University, Seoul, Republic of Korea   [3]Department of Cell and Chemical Biology, Leiden University Medical Centre, Leiden, The Netherlands

Correspondence: donghyuk.shin@yonsei.ac.kr

The ubiquitination system is exclusive to eukaryotes; thus, invading bacteria and viruses have evolved survival strategies to hijack the host ubiquitination system during infection (Lin & Machner, 2017; Wimmer & Schreiner, 2015), as their genomes encode ubiquitin ligases or DUBs that redirect or eliminate host signals. For example, *Salmonella typhimurium* contains the HECT-type E3 ligase SopA (Diao et al, 2008; Fiskin et al, 2017) and *Legionella pneumophila* contains LegU1 and LubX, which are similar to the F-box– and U-box–containing E3 ligases, respectively (Kubori et al, 2008; Ensminger & Isberg, 2010; Quaile et al, 2015). Not only E3 ligases but also pathogen-encoded DUBs can influence host cellular processes, such as immune response, autophagy, and morphology (Mesquita et al, 2012; Pruneda et al, 2018; Wan et al, 2019).

*L. pneumophila*, the causative agent of Legionnaire's disease, has developed numerous mechanisms of hijacking or inhibiting host ubiquitin signaling. Through the defective organelle trafficking/intracellular multiplication (Dot/Icm) type IV secretion system, *L. pneumophila* secretes more than 300 effector proteins into the cytoplasm of the host cell, thereby regulating among others host ubiquitin signals (Segal et al, 2005; Qiu & Luo, 2017). A representative example is the SidE ligase family (SdeA, SdeB, SdeC, and SidC) of *Legionella* that mediate an unconventional phosphoribosyl (PR) serine ubiquitination system (Bhogaraju et al, 2016; Qiu et al, 2016; Kalayil et al, 2018; Bhogaraju et al, 2019; Black et al, 2019; Shin et al, 2020b). More recently, several studies identified human OTU-like DUBs from *Legionella* and named them *Legionella* OTU-like DUBs (Lot). Lots include LotA (Lpg2248/Lem21), LotB (Lpg1621/Ceg23), LotC (Lpg2529/Lem27), and LotD (Lpg0227/Ceg7) (Ashida et al, 2014; Kubori et al, 2018; Shin et al, 2020a). Similar to human OTUs, Lot DUBs exhibit linkage specificity. LotB and LotC prefer the K63 and K48 linkage, respectively. In contrast, LotD cleaves K6, K11, K33, K48, and K63 linkage (Shin et al, 2020a; Kitao et al, 2020; Schubert et al, 2020). Interestingly, LotA comprises two OTU-like domains with two catalytic cysteine residues (C13 and C303). The first OTU domain of LotA (LotA OTU1) exclusively cleaves the K6 linkage, whereas the second OTU domain (LotA OTU2) cleaves K48 and K63 linkages (Kubori et al, 2018; Takekawa et al, 2022).

Despite various efforts, however, the conserved molecular mechanism of these unique pathogenic DUBs has not been clearly understood. In this study, therefore, we investigated the structure and molecular mechanism of Lots. We found that Lots share a unique structural topology that is not found in any other OTUs. In addition, the catalytic triads of Lots were revealed to have differences compared with other OTUs. Moreover, we also provide molecular insights into how Lots distinguish specific Ub linkage types and length of ubiquitin chains.

## Results

### LotA domains exhibit cooperative catalysis

*Legionella* contains four OTU-DUBs (LotA [Lpg2248/Lem21], LotB [Lpg1621/Ceg23], LotC [Lpg2529/Lem27], and LotD [Lpg0227/Ceg7]) that exhibit distinct ubiquitin chain specificity. Among them, LotA comprises two OTU domains (LotA OTU1 and LotA OTU2; Fig 1A).

Previous reports defined the linkage specificity of each OTU domain of LotA (Kubori et al, 2018; Takekawa et al, 2022). LotA OTU1 preferentially cleaves K6-linked chains, whereas LotA OTU2 cleaves K48 and K63 chains. Consistent with the previous reports, the LotA OTU1 domain (LotA$_{7–290}$) effectively cleaved K6-linked diubiquitin chains (Fig 1B). However, neither LotA OTU2 (LotA$_{294–544}$) nor LotA OTU1_OTU2 (LotA$_{7–544}$) domains cleaved K48- or K63-linked diubiquitin chains (Fig 1C and D). We further analyzed the K48 or K63 linkage specificity of LotA OTU2 with diubiquitin activity-based probes (Fig S1A–D). Propargyl (Prg) probes, which have a Prg group at the C-terminus of the diubiquitin chain, can detect S1 ubiquitin-binding site (active site where a distal ubiquitin binds) and S2 (third ubiquitin-binding site on the DUBs) sites of DUBs (Ekkebus et al, 2013; Sommer et al, 2013; Flierman et al, 2016). Vinyl amide (VA) probes, which have a VA reactive group between the two ubiquitin moieties of a diubiquitin chain, were developed to identify S1 and S1′ (binding region for proximal ubiquitin in a polyubiquitin chain) sites on DUBs (Mulder et al, 2014). Similar to the results obtained for the cleavage analysis of diubiquitin chains, none of the LotA OTU domain constructs (OTU1, OTU2, and OTU1_OTU2) reacted with the diubiquitin activity-based probes of K48- or K63-linked chains (Fig S1A–D), whereas monoubiquitin Prg probes react with all the constructs (Fig S1E). These results indicate that the catalytic cysteines of OTU1 (C13) and OTU2 (C303) domains did not meet and react with the chemical warheads from diubiquitin probes, whereas the propargyl warhead from monoubiquitin is freely accessible to both catalytic cysteines.

Interestingly, all previous studies used longer polyubiquitin chains with more than two ubiquitin residues to elucidate the cleavage of K48- and K63-linked ubiquitin chains by the OTU2 domain (Kubori et al, 2018; Takekawa et al, 2022). Therefore, we examined the catalytic activity of the LotA OTU2 domain in cleaving K48- or K63-linked tetraubiquitin chains. We determined that the LotA OTU2 domain cleaved both K48- and K63-linked tetraubiquitin chains, whereas the LotA OTU1 domain did not cleave either of the two chains (Fig 2A–C). These results revealed the length-dependent catalytic activity of LotA OTU2, which only cleaves polyubiquitin chains with more than two ubiquitin moieties. In contrast, the K6 linkage specificity of LotA OTU1 was not restricted by the length of polyubiquitin chains, as it cleaved a K6-linked diubiquitin chain (Fig 1B).

Surprisingly, the cleavage of K48- and K63-linked tetraubiquitin chains by LotA was remarkably enhanced when intact OTU1 and OTU2 domains (LotA$_{7–544}$) were used for the cleavage assay (Fig 2B and C). Although the apparent processing speed of K48-linked chains is significantly higher than K63-linked chains, both K48- and K63-linked chains were cleaved much more with the OTU1_OTU2 construct than the OTU2 construct, indicating the supportive role of the OTU1 domain in K48- or K63-linked chain processing. Subsequently, we investigated whether the catalytic activity of the OTU1 domain was required for the cleavage of K48- and K63-linked chains by the OTU2 domain. Thus, we mutated the catalytic cysteine (Cys13) in OTU1 to alanine (LotA OTU1*-OTU2; LotA$_{7–544}$ C13A) and observed that the LotA OTU1*-OTU2 construct retained its catalytic activity and cleaved a K48-linked tetraubiquitin chain at the same rate as that of the wild type (WT; Fig 2D), indicating that the catalytic activity of the OTU1 domain is not

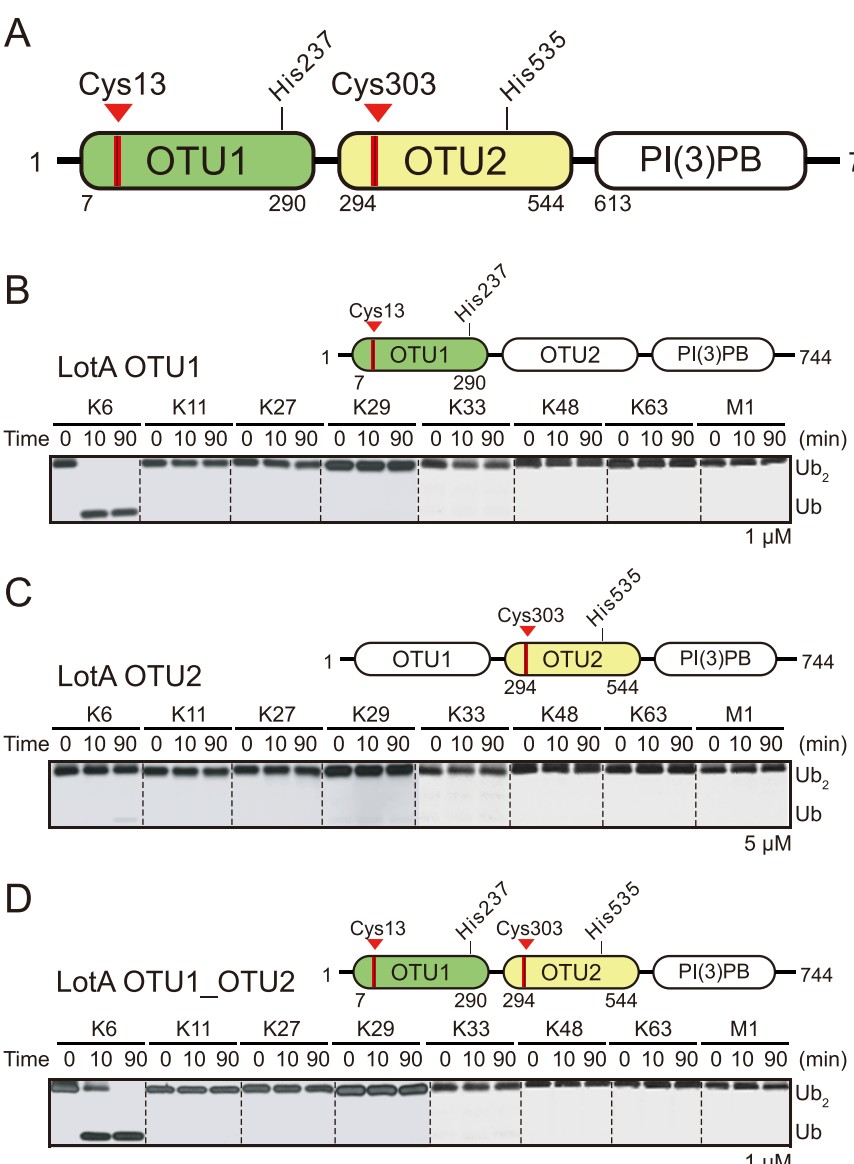

Figure 1. Deubiquitinase activities of both OTU domains of LotA.
**(A)** Schematics of *L. pneumophila* LotA domain architecture. LotA comprises two OTU domains (LotA OTU1 and LotA OTU2). Catalytic cysteines of each OTU domain are labeled. **(B)** Diubiquitin cleavage assay by LotA OTU1 (LotA$_{7–290}$) against eight different diubiquitin linkages. Reactions were quenched at the indicated time-points and analyzed on SDS–PAGE with silver staining. **(C)** Diubiquitin cleavage assay by LotA OTU2 (LotA$_{294–544}$) against eight different diubiquitin linkages. Reactions were quenched at the indicated time-points and analyzed on SDS–PAGE with silver staining. **(D)** Diubiquitin cleavage assay by LotA OTU1_OTU2 (LotA$_{7–544}$) against eight different diubiquitin linkages. Reactions were quenched at the indicated time-points and analyzed on SDS–PAGE with silver staining.
Source data are available for this figure.

necessary for the cleavage of the K48- or K63-linked polyubiquitin chains, but its presence is crucial for the catalytic activity of the OTU2 domain (Fig 2D and E).

Next, to understand the molecular mechanism underlying the interplay between the OTU1 and OTU2 domains, we analyzed the overall architecture of LotA OTU1_OTU2 (LotA$_{7–544}$) using electron microscopy. Even though the protein size of LotA$_{7–544}$ is on the border of a current cryo-EM technique, we expected the observation of the relative orientation and overall shape. 2D class averages of the negative-stain and cryo-images displayed heterogeneous classes with two distinct globular domains (Figs 3A and S2A). We then reconstructed a cryo-EM map at ~11 Å (cryo-EM; Figs 3B and E and S2B), showing two converged densities of the LotA$_{7–544}$. Although the resolution was not sufficient to model the precise atomic details of both OTU domains, the overall dimensions suggested that the two OTU domains were not separated but

aligned in proximity (Fig 3B and Table S1). Although our cryo-EM work was underway, a crystal structure of LotA OTU1_OTU2 was deposited (PDB: 7W54). Intriguingly, the crystal structure contains completely different conformations of LotA molecules in the asymmetric unit (ASU). Superimposition of the OTU1 domain of both conformers in the ASU exhibited ~180° rotation of the OTU2 domain, indicating that the relative orientation of the OTU1 and OTU2 domains varied in the solution (Fig 3C). We could not fit both high-resolution conformers into our cryo-EM electron density map. Therefore, we hypothesized that the cryo-EM electron density map presented the different orientation of OTU1-OTU2 domains with respect to each other and perhaps the domain orientation is variable in solution. These results prompted us to assume that a specific structural arrangement of the OTU1 and OTU2 domains might be required for the recognition and cleavage of the K48- and K63-linked tetraubiquitin chains (Fig 2B–D). To prove that the OTU1

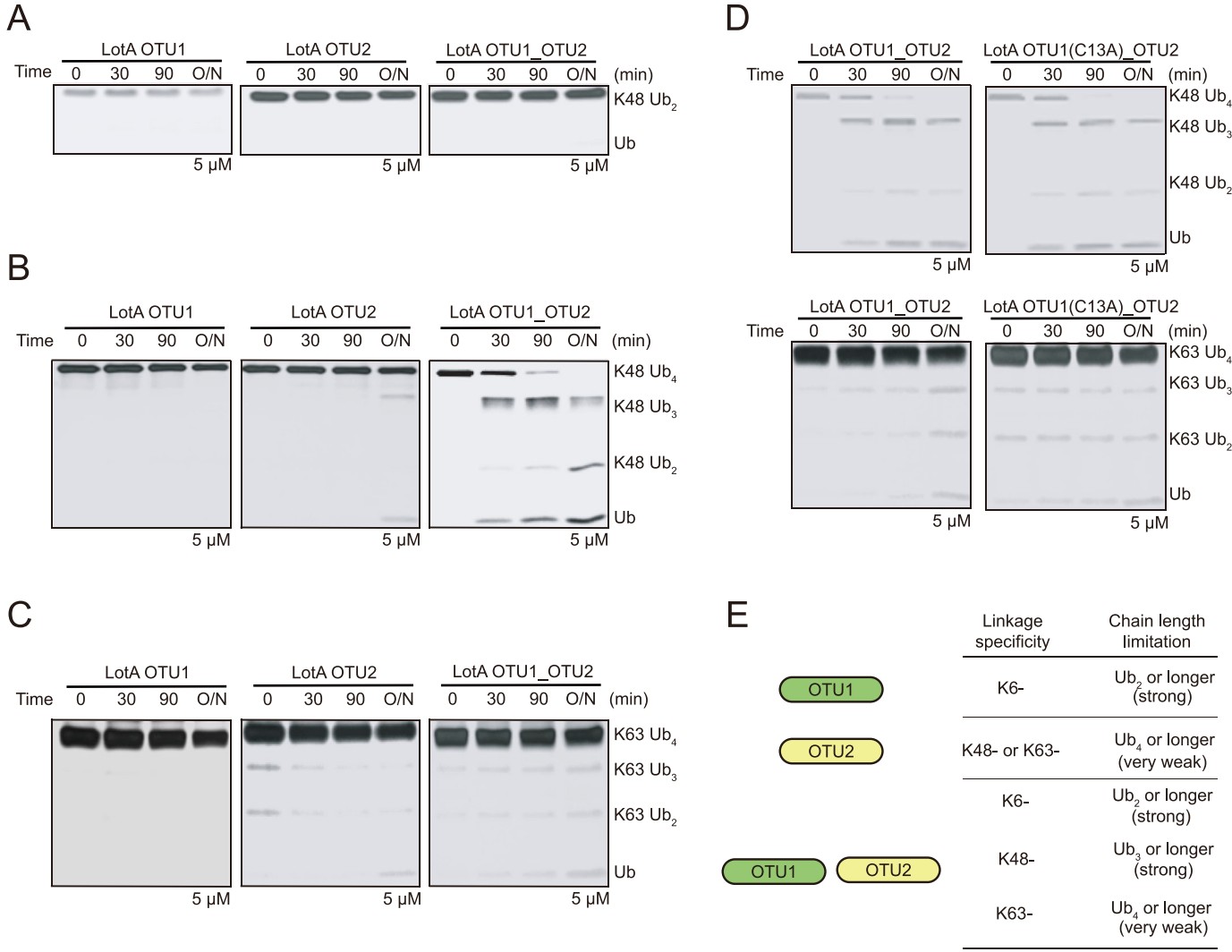

**Figure 2. Cooperative catalytic activity of LotA OTU1 and OTU2 domains.**
**(A)** K48-linked diubiquitin cleavage assay by LotA OTU constructs (OTU1, OTU2, and OTU1_OTU2). **(B)** K48-linked tetraubiquitin chain cleavage assay by LotA OTU constructs (OTU1, OTU2, and OTU1_OTU2). Catalytic activity against K48-linked tetraubiquitin chains was remarkably enhanced when intact OTU1 and OTU2 domains (LotA$_{7-544}$) were used. **(C)** K63-linked tetraubiquitin chain cleavage assay by LotA OTU constructs (OTU1, OTU2, and OTU1_OTU2). Catalytic activity against K63-linked tetraubiquitin chains was enhanced when intact OTU1 and OTU2 domains (LotA$_{7-544}$) were used. **(D)** K48-linked tetraubiquitin chain cleavage assay by WT LotA OTU1_OTU2 and LotA OTU1*_OTU2 (C13A) mutant. **(E)** Ubiquitin linkage and length preference of LotA OTU constructs (OTU1, OTU2, and OTU1_OTU2).
Source data are available for this figure.

domain structurally assists the OTU2 domain, we deleted a linker region (276–283) between the two OTU domains and restricted the conformational spaces of the OTU domains. We observed that the linker deletion mutants exhibited slower cleavage of K48-linked tetraubiquitin chains than the WT or C13A mutants (Fig 3D and E), indicating that structural flexibility between the two domains is required for K48-linked tetraubiquitin recognition and cleavage. We also examined K63-linked tetraubiquitin chain cleavage with the deletion mutant (Fig S2C and D). In contrast to K48-linked chains, the deletion mutant did not significantly affect the cleavage rate of K63-linked tetraubiquitin chains. Together, our results reveal that the OTU1 domain of LotA specifically cleaves K6-linked polyubiquitin chains without any length limitation, whereas the OTU2 domain processes K48- or K63-linked polyubiquitin chains in the

presence of more than two ubiquitin moieties. Moreover, our data revealed that the OTU1 domain is not only required for K6-linked polyubiquitin chain cleavage but also assists the OTU2 domain in recognizing and cleaving K48- or K63-linked polyubiquitin chains.

## Crystal structure of the OTU1 domain of LotA

To elucidate the mechanism underlying the cleavage of K6-linked polyubiquitin chains by the LotA OTU1 domain, we successfully obtained the crystal structure of the LotA OTU1 domain. Based on secondary structural analysis, we designed a LotA OTU1 construct spanning residues 7–290 (LotA$_{7-290}$). The LotA$_{7-290}$ crystal diffracted at 1.54 Å resolution, and we determined the structure by molecular replacement using the structure predicted by the AlphaFold Protein

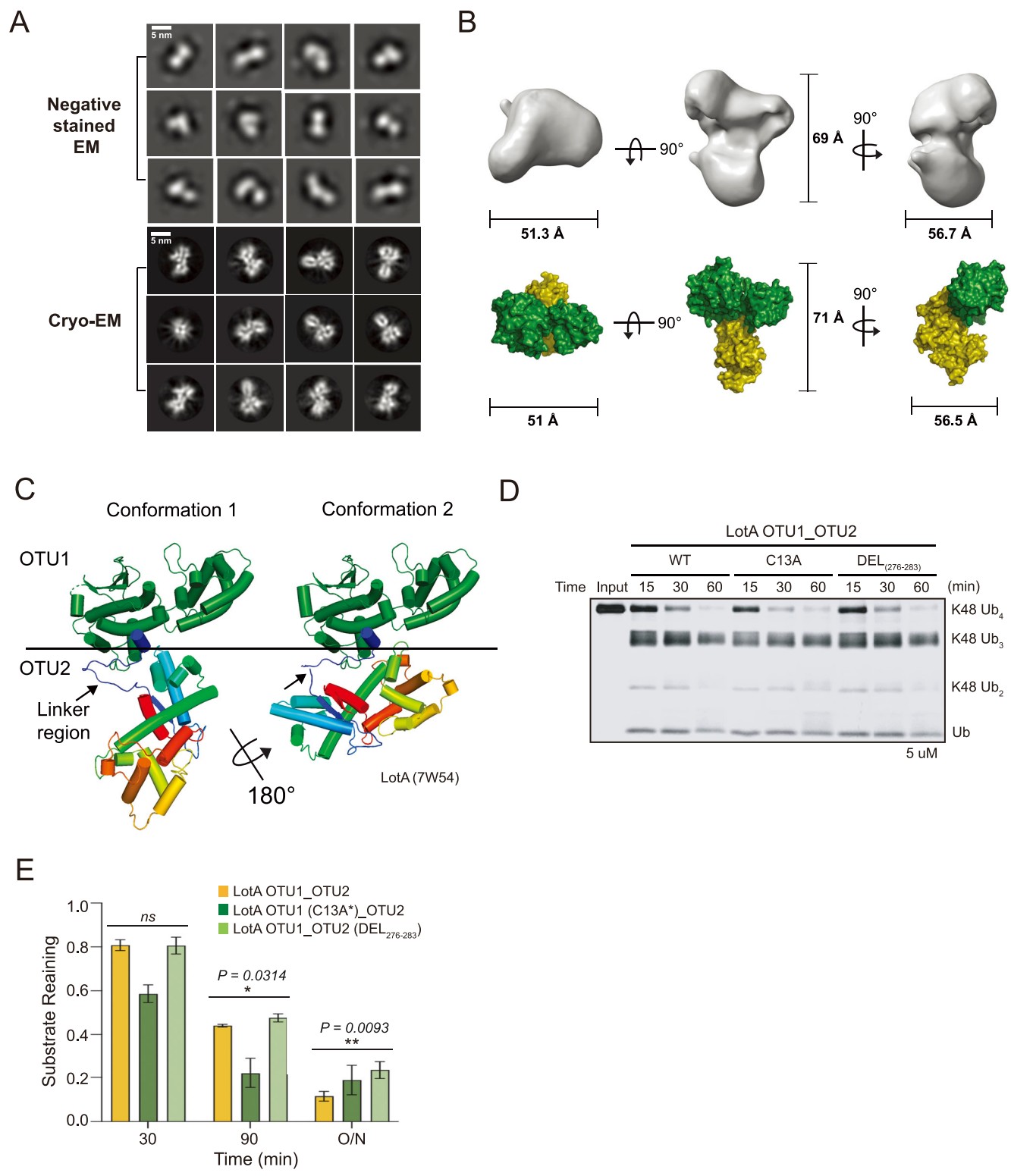

**Figure 3.  Structure of LotA OTU1_OTU2.**
**(A)** Selected 2D classes of negative-stain (upper) and cryo-EM (down) LotA OTU1_OTU2. Scale bar represented 5 nm. **(B)** LotA$_{7-544}$ cryo-EM model orientated in X, Y, and Z axis. Each distance is described as angstrom (Å) (upper). Proposed OTU1 and OTU2 orientation of LotA$_{7-544}$ (down) using crystal structure (PDB: 7W54). **(C)** Crystal structure of LotA OTU1_OTU2 (7W54). Two completely different conformations of LotA molecules are observed in the asymmetric unit. **(D)** DUB assay comparing cleavage of K48-linked tetraubiquitin chains by wild-type LotA OTU1_OTU2, LotA OTU1*_OTU2 (C13A) mutant, and linker deletion (residues 276–283) mutant of LotA OTU1_OTU2.
**(D, E)** Quantification of remaining K48-linked tetraubiquitin chains shown in (D). Data are presented as the mean ± S.D (n = 3, * 0.01 < $P$ < 0.05 and ** 0.001 < $P$ < 0.01). Source data are available for this figure.

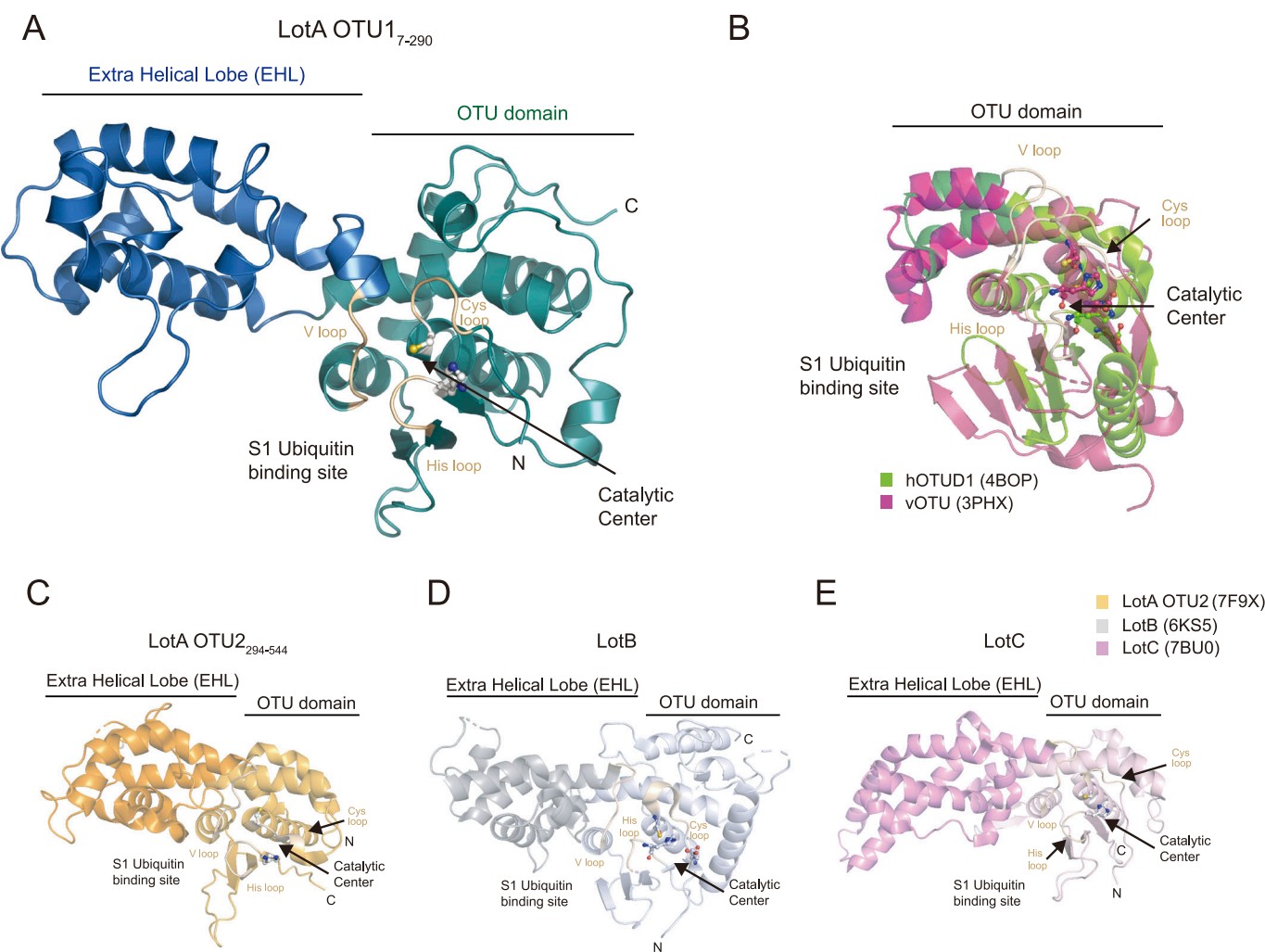

**Figure 4. Structural comparison of *Legionella* OTU deubiquitinases.**
**(A)** X-ray crystal structure of LotA OTU1 (LotA$_{7-290}$) diffracted to 1.54 Å determined by molecular replacement (deposited PDB ID: 8GOK). Catalytic cysteine and histidine of LotA OTU1 are shown as a ball-and-stick model, and Cys loop, His loop, and V loop are highlighted in wheat color. The EHL domain and the catalytic OTU domain are labeled. **(B)** Structural comparison of hOTUD1 (4BOP, green) and vOTU (3PHX, pink). Structures were aligned by their core OTU domain. **(C)** Crystal structure of LotA OTU2 (7F9X). Catalytic cysteine and histidine of LotA OTU2 are shown as a ball-and-stick model. The EHL domain and the catalytic OTU domain are labeled. **(D)** Crystal structure of LotB (6KS5). Catalytic triad residues of LotB are shown as a ball-and-stick model. The EHL domain and the catalytic OTU domain are labeled. **(E)** Crystal structure of LotC (7BU0). Catalytic triad residues of LotC are shown as a ball-and-stick model. The EHL domain and the catalytic OTU domain are labeled.

Structure Database as a template model (Figs 4A and S3A) (Evans & McCoy, 2008; DiMaio et al, 2011; Scapin, 2013; Jumper et al, 2021). The crystal belonged to the tetragonal space group P4$_1$2$_1$2, with one molecule per ASU. The final refined model of LotA$_{7-290}$ contained 270 residues of LotA (Ala7 to Thr276) and three amino acids from the GST expression tag (Table 1).

The overall structure of LotA$_{7-290}$ shows a dumbbell-like fold harboring a typical OTU fold (residues 7–90 and 200–276) and an EHL part (residues 91–199) (Figs 4A and B and S4A) (Mevissen et al, 2013; Shin et al, 2020a). The OTU domain of LotA$_{7-290}$ comprises eight α-helices (α1–α5 and α13–α15) and a β-sheet with three β-strands (β1–β3). The catalytic center of the OTU domain is composed of a catalytic cysteine residue (Cys13) residing on α1 (residues 13–27) and a β-sheet containing the catalytic histidine residue (H237). The EHL of the LotA OTU1 domain is composed of only α-helices and is located before

the conserved V loop and near the S1 ubiquitin-binding site, suggesting a putative role of the EHL in K6-linked polyubiquitin selectivity of LotA OTU1 (Fig 4A and C–E) (Kubori et al, 2018; Shin et al, 2020a). To examine whether the extra insertion of the EHL domain of LotA$_{7-290}$ resembled any known structures, we performed a structural similarity analysis. The Dali structural similarity analysis server found that the structure of LotA$_{7-290}$ resembles previously known Lot structures (LotA_OTU2 [PDB: 7F9X], LotB [PDB: 6KS5], and LotC [PDB: 7BU0]), indicating that the insertion of the EHL domain is a unique structural feature of the Lot family and distinguishes it from other DUB families.

To further characterize the EHL domain of the Lot DUB family, we compared the topology of the EHL domains of all known Lot DUBs (Kubori et al, 2018; Shin et al, 2020a; Liu et al, 2020; Ma et al, 2020; Takekawa et al, 2022). Our analysis revealed that the EHL domains of all Lot family members have similar topologies (Fig 5A and B). The EHL

**Table 1. X-ray data collection and refinement statistics.**

| | LotA$_{7-290}$ |
|---|---|
| Data collection | |
| Space group | P 41 21 2 |
| Cell dimensions | |
| a, b, c (Å) | 86.217 86.217 91.701 |
| α, β, γ (°) | 90, 90, 90 |
| $R_{merge}$ | 0.01733 (0.4571) |
| CC$_{1/2}$ | 1 (0.641) |
| CC* | 1 (0.884) |
| I/σI | 18.91 (1.56) |
| Completeness | 99.96 (99.94) |
| Redundancy | 2.0 (2.0) |
| Refinement | |
| Resolution (Å) | 36.64–1.52 (1.574–1.52) |
| No. of reflections | 107,431 (10,592) |
| $R_{work}/R_{free}$ | 0.1977/0.2087 |
| No. of atoms | 2,559 |
| Macromolecules | 2,236 |
| Solvent | 323 |
| B-factors | |
| Macromolecules | 26.91 |
| Solvent | 36.74 |
| R.m.s deviations | |
| Bond lengths (Å) | 0.007 |
| Bond angles (°) | 0.93 |

Statistics in parentheses are for the highest resolution shell.

domain contains a roof-like fold, which is composed of three helices before the V loop, and an amphipathic alpha helix is present below the roof folding. The hydrophobic region of the amphipathic helix is in contact with the roof helices, whereas the hydrophilic surface is exposed to the surface. To validate whether this structural feature of the EHL was also present in LotD (Ceg7), the AlphaFold-predicted structure of LotD was analyzed (Fig S4B and C). Interestingly, the EHL of LotD has the same topology as that of the other Lots. To further validate whether the EHL fold is found only in the Lots, we performed a structural–fold similarity analysis of the EHL. The entire AlphaFold-predicted protein structures are compared with the EHL domain of LotA through the Foldseek search server (van Kempen et al, 2022 *Preprint*). The analysis only picked LotA orthologues from other *Legionella* species (*Legionella antarctica* and *Legionella moravica*). These results support that the topology of the EHL is unique to the entire Lot DUB family and can be considered a standard characteristic of Lots.

### Conformational change in the EHL is required for K6-linked polyubiquitin chain cleavage by LotA

During the structure determination of the LotA$_{7-290}$ with molecular replacement using the AlphaFold-predicted structure as a template

model, we could not obtain a successful solution. However, when the OTU and EHL domains were manually divided and used as two individual templates, we obtained the phase information and successfully determined the structure of LotA OTU1$_{7-290}$ (Fig S3A). This led us to explore the conformational diversity of the EHL of the OTU1 domain. Moreover, previous studies on LotC have revealed that binding of ubiquitin to OTU induces conformational changes in the EHL domain (Fig S3B) (Shin et al, 2020a; Liu et al, 2020). To examine whether this ubiquitin binding–dependent conformational change was also required for the catalytic activity of LotA OTU1, we designed an experiment to restrict the conformational flexibility of the EHL domain. Based on the structural comparison, four amino acids (N190 to A193) in the hinge region between the EHL and the OTU domains were chosen for examination. We assumed that substitution of each of these amino acids with proline restricted the relative movement of the EHL to the OTU domain. Indeed, the cleavage of a K6-linked diubiquitin chain was completely abolished in all three proline mutants (D190P, H192P, and A193P) (Fig 5C–E). We also introduced an alanine mutation at P191 in the hinge region to examine whether it induces the EHL domain to a specific orientation with respect to the OTU1 catalytic domain. However, the catalytic activity of the P191A mutant was similar to that of the wild-type OTU1, suggesting that P191 was not responsible for placing the EHL in a specific orientation with respect to the OTU domain. Thus, our results suggest that the conformational flexibility of the EHL domain of OTU1 is crucial for K6-linked diubiquitin chain cleavage, and this conformational change in the EHL domain upon recognition of the ubiquitin moiety could be a unique characteristic of the Lot DUB family.

### Catalytic triad of lot DUBs resembles that of the A20 OTU family

The catalytic sites of cysteine proteases, including OTU-DUBs, harbor three residues (catalytic triads, cysteine, histidine, and an acidic amino acid; Asp or Glu) (Baker et al, 1993; Buller & Townsend, 2013; Mevissen et al, 2013; Verma et al, 2016). Both cysteine and histidine residues actively participate in cleaving the peptide bonds. The catalytic cysteine residue attacks the carbonyl carbon of the scissile peptide bond, and this step requires a histidine residue that accepts a hydrogen from the thiol group of the cysteine residue. The third residue of the catalytic triad is acidic (aspartic acid or glutamic acid), and its primary role is to stabilize the charge on the imidazole ring of the histidine residue during catalysis. Although both cysteine and histidine residues are indispensable for peptide bond cleavage and are highly conserved in the cysteine protease family, the acidic residue is not required for the activity of some cysteine proteases, which have catalytic dyads (Nakagawa, 2013; Elsasser et al, 2017; Ramos-Guzman et al, 2020; Ferreira et al, 2021). Recent studies on DUBs from different bacteria have suggested variations in the catalytic triad (Akutsu et al, 2011; Nakagawa, 2013; Pruneda et al, 2016; Shin et al, 2020b; Schubert et al, 2020; Takekawa et al, 2022). However, the catalytic residues of Lot DUBs have not been clearly defined. For instance, although both LotB and LotC possess an acidic amino acid residue (D21 and D17, respectively) present N-terminally from the catalytic cysteine residue (C29 and C24, respectively), LotA_OTU2 and LotD have an asparagine or a

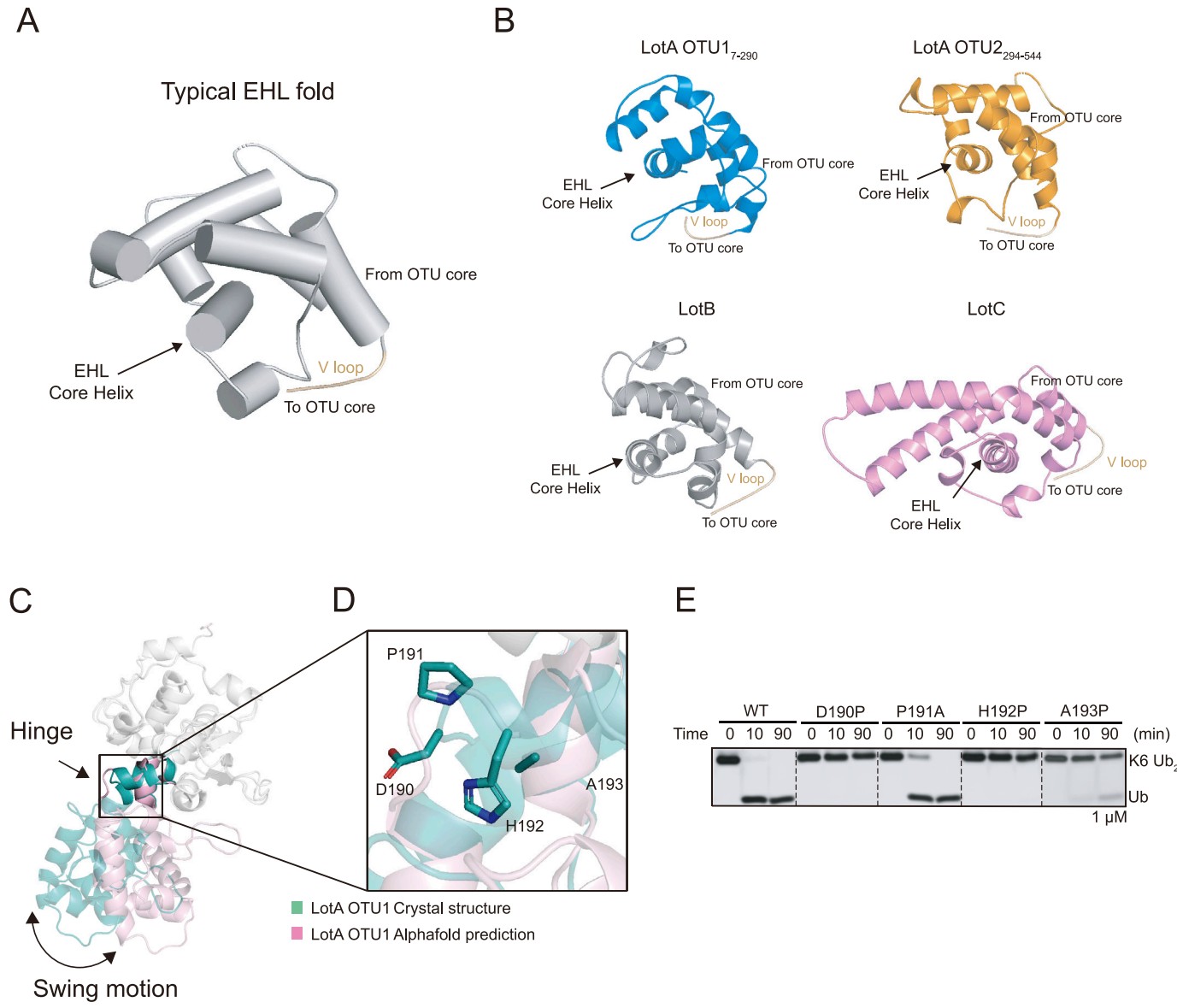

**Figure 5. Structural analysis of the EHL domain of Lots.**
**(A)** Typical EHL fold architecture of the LOT DUB family. **(B)** Structure of the EHL domain of the LOT DUB family. The EHL domain of the LOT DUB family shared roof-like folding. **(C)** Structural comparison of LotA OTU1 crystal structure (cyan) and LotA OTU1 AlphaFold prediction (pink). The OTU domain of two structures is superimposed, and relative orientation of the EHL domain to the OTU domain is presented. **(D)** Close-up view of hinge region of LotA OTU1. Four amino acids (residues 190–193) in the hinge region between EHL and OTU domains are shown as a stick model. **(E)** DUB assay against K6-linked diubiquitin with wild-type and the hinge mutants of LotA OTU. Source data are available for this figure.

threonine residue present C-terminally from the catalytic cysteine residues.

To elucidate the catalytic residues of Lot DUBs, we analyzed the structures of Lot DUBs. Because misaligned catalytic triads are often observed in the crystal structures, we also compared the structures predicted using the AlphaFold database (Fig 6A). Previous studies have reported catalytic triads of both LotB and LotC, where the arrangement of triads resembles the catalytic site of A20 or Cezanne-type OTUs (Komander & Barford, 2008; Mevissen et al, 2016; Shin et al, 2020a; Ma et al, 2020). Both LotB and LotC have an acidic residue located N-terminally from the catalytic cysteine

residue, which differs from typical OTU-DUBs, which have an acidic residue located two amino acids C-terminal of the histidine residue on the same beta-strand (His-XX-Acidic residue; Fig 6B, E, and F). Interestingly, similar to the structures of LotB and LotC, our structural analysis revealed the presence of negatively charged residues toward the N-terminal region from the catalytic cysteine residue in case of other Lot DUBs, including LotA_OTU2 and LotD. Both LotA_OTU2 and LotD exhibited a well-positioned catalytic triad with the histidine base between the catalytic cysteine residue and acidic amino acids (Fig 6A). Consistent with the structural analysis, mutation at D296 (D296A) of LotA OTU2 and D6 (D6A) of LotD

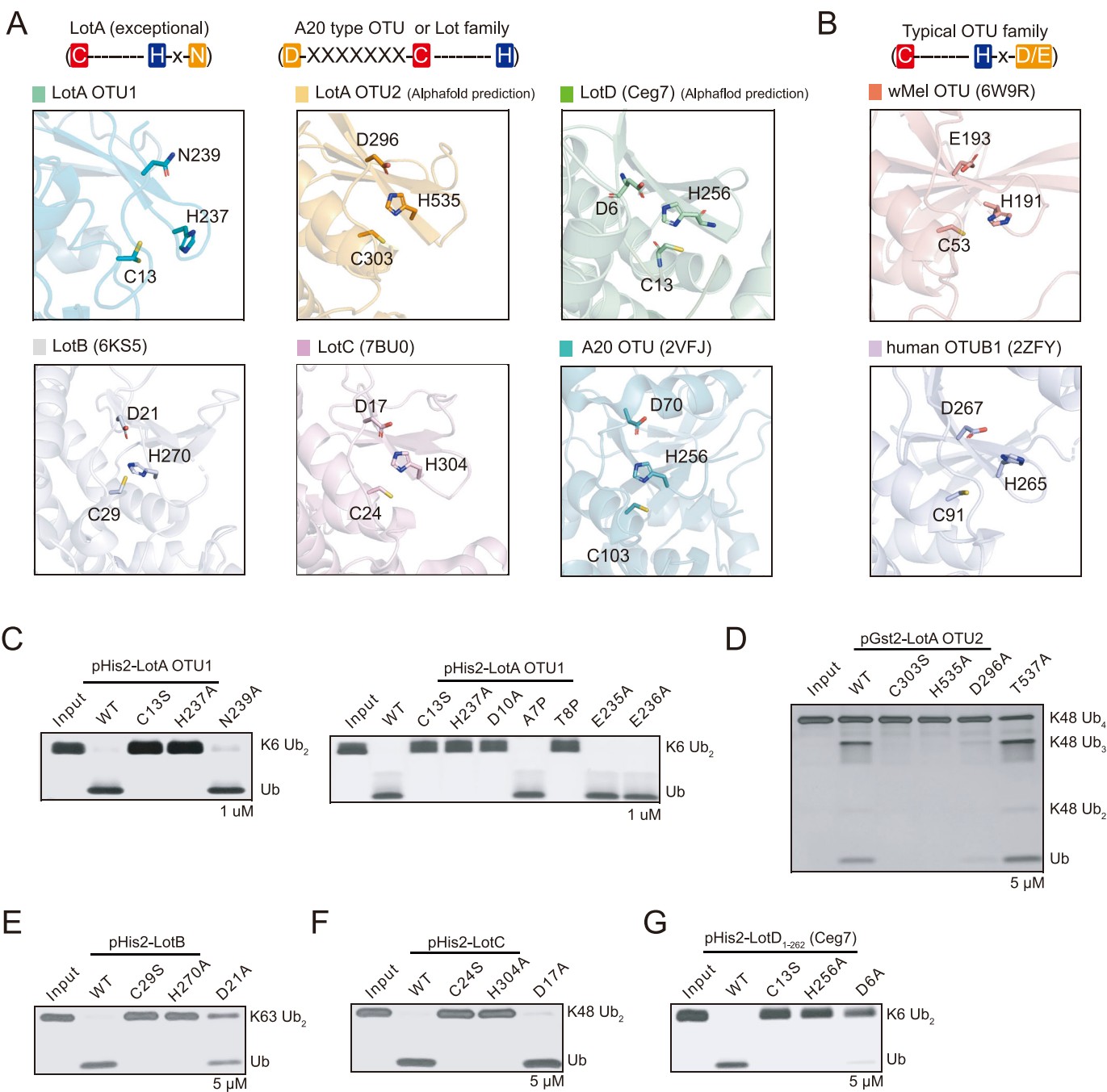

**Figure 6. Catalytic triad of the LOT DUB family.**
**(A)** Close-up views of the LotA OTU1, LotA OTU2, LotB (6KS5), LotC (7BU0), LotD, and A20 OTU (2VFJ) catalytic sites. Catalytic triads are highlighted as a stick model. Schematic representation of the order of catalytic triads is also presented. **(B)** Close-up views of the wMel OTU (6W9R) and hOTUB1 (2ZFY) catalytic sites. Catalytic triads are highlighted as a stick model. Schematic representation of the order of catalytic triads is also presented. **(C, D, E, F, G)** Diubiquitin or polyubiquitin cleavage assay by the LOT DUB family to define catalytic triad residues. **(C, D, E, F, G)** LotA-OTU1, (D) LotA-OTU2, (E) LotB, (F) LotC, and (G) LotD. The catalytic activity of the LOT DUB family WT and their mutants was tested. Reactions were quenched at the indicated time-points and resolved by SDS–PAGE with silver staining, respectively.
Source data are available for this figure.

reduced the enzymatic activity of LotA_OTU2 and LotD, respectively (Fig 6D and G). Previous reports have suggested that both LotA OTU2 have a threonine residue at the C-terminal position to the base histidine residue, which is similar to canonical OTU-DUBs

(Edelmann et al, 2009; Mevissen et al, 2013; Clague et al, 2019; Schubert et al, 2020; Takekawa et al, 2022). However, the mutation at the T537 position of LotA OTU2 (T537A) did not affect the catalytic activity, suggesting that D296 is N-terminally from the catalytic C303

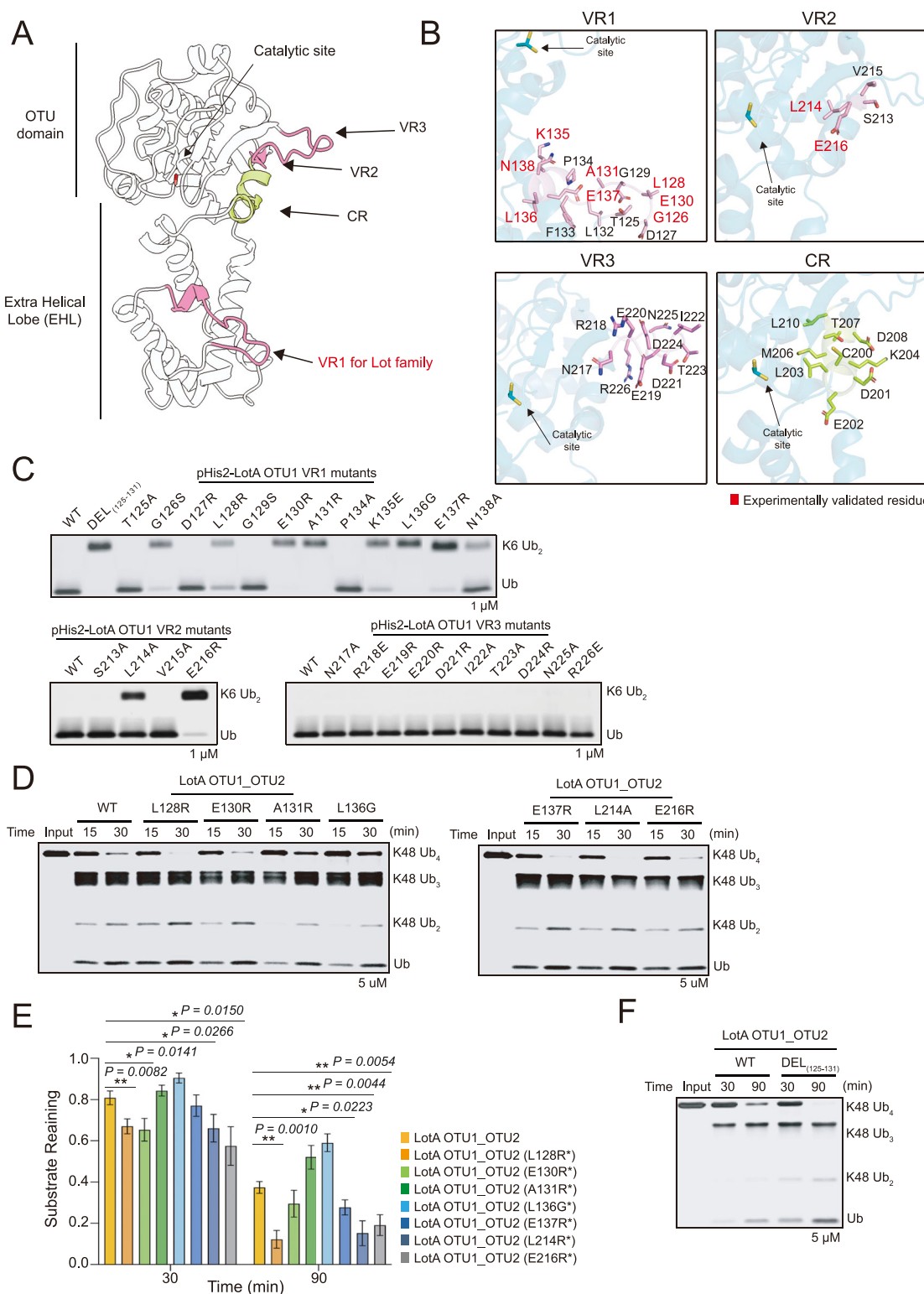

**Experimentally validated residues**

**Figure 7. S1 ubiquitin-binding site of LotA OTU1.**
**(A)** Cartoon representation of LotA OTU1. The constant region (CR, green) and three variable regions (VRs, pink) are the parts of the S1 ubiquitin-binding site. The EHL domain contains the VR1 region of the LOT DUB family. **(B)** Close-up views of the VRs and CR of LotA OTU1. Three VRs (pink) and CR (green) residues are represented as a stick model, and experimentally identified residues are highlighted in red color. **(C)** K6-linked diubiquitin cleavage assay by LotA OTU1 WT and its mutants or deletion of VRs (VR1, VR2, and VR3) in LotA OTU1. Reactions were quenched at the indicated time-points and resolved by SDS–PAGE with silver staining. **(D)** Polyubiquitin cleavage assay by LotA OTU1_OTU2 WT and VR1 mutants of LotA OTU1 as labeled. **(D, E)** Quantification of remaining K48-linked tetraubiquitin chains shown in (D). Data are

residue of LotA OTU2 as the acidic residue complementing the catalytic triad of LotA OTU2. These results indicate that the catalytic triads of Lot DUBs belong to the A20 type, which has an acidic amino acid present located N-terminally from the catalytic cysteine residue. We also validated other predicted residues of LotA_OTU2 and LotD, which also comprised catalytic triads, by introducing mutations in the predicted residues (Figs 6D and G and S5). All mutations, indeed, completely abolished the catalytic activity of Lot DUBs. For LotA OTU1, we could not identify the acidic residue present N-terminally from the catalytic cysteine residue. Instead, LotA OTU1 had an asparagine residue (N239) at the C-terminus of the base histidine residue (H237). Mutation N239 (N239A) exhibited less catalytic activity than the WT, and more K6-linked diubiquitin chains were present (Fig 6C), indicating that the catalytic core of LotA OTU1 follows that of typical OTU-DUBs. Further structural analysis of LotA OTU1 revealed two glutamate residues (E235 and E236) located at the N-terminal from the H237 residue and an N239 residue at the two amino acids C-termini of H237. However, mutations in these residues did not affect the catalytic activity. Instead, D10, which is located toward the N-terminus from the catalytic cysteine residue, completely abolished the cleavage of K6-linked diubiquitin chains. Because both the crystal and AlphaFold-predicted structures did not exhibit the possibility of the presence of D10 to stabilize H237, and D10 is a part of the oxyanion hole, we assumed that the D10A mutant altered the conformation of the oxyanion hole loop. To test this hypothesis, we introduced a proline mutation close to (T8P) or further away (A7P) from Asp10. Interestingly, the single proline mutation at the T8 position also completely diminished the catalytic activity of the LotA, supporting the role of D10 in the formation of an oxyanion hole. Together, these results revealed that Lot DUBs resemble the catalytic triad of A20 OTU, although LotA OTU1 possesses a slightly altered catalytic core compared with other OTU-DUBs.

### Structural elements of the S1 ubiquitin-binding site of the Lot family

Previous studies on bacterial DUBs, including CE-clan ubiquitin or ubiquitin-like proteases and OTU-DUBs, have reported the key regions that form the S1 ubiquitin-binding site (Akutsu et al, 2011; James et al, 2011; Mevissen et al, 2013; Pruneda et al, 2016; Schubert et al, 2020). OTU-DUBs include a structurally conserved helix (constant region, CR) and three variable regions (VRs, VR1–3) (Schubert et al, 2020). In general, VR1 is a helical arm located underneath the CR helix, VR2 refers to the edge of the beta-sheets, and VR3 is the beta-hairpin region. Because all Lot family members have an EHL domain, which is located close to the S1 ubiquitin-binding site, we investigated the effects of the EHL domain on the S1 ubiquitin-binding site. Structural comparison of Lot DUBs revealed that the position of the Lot EHL domain overlapped with the VR1 of other OTU-DUBs (Fig S6A). Moreover, previous studies on LotB and LotC have experimentally verified several residues that are crucial

for ubiquitin binding (Fig S6C–E), and these residues are concentrated in the C-terminal helical loop region of the EHL core helix. Interestingly, this helical loop region is also conserved in the crystal structures of LotA OTU1 and OTU2 domains and the predicted structure of LotD, suggesting their putative role in ubiquitin binding (Figs S4C and S6B). Based on these observations, we hypothesized that this helical loop region in the EHL domain forms the VR1 in the Lot family (Fig 7A). To test this, we mutated a number of residues in this loop of LotA OTU1 and examined the catalytic activity of these mutants. Surprisingly, eight of the 12 single amino acid mutants were considerably less active in cleaving K6-linked diubiquitin chains compared with the WT (Fig 7B and C), implicating the role of this helical loop as VR1 to form the S1 ubiquitin-binding site in the Lot family. To further characterize the S1 ubiquitin-binding site, we examined other VRs (VR2 and VR3) of LotA OTU1. Two mutations (L214A and E216R) in VR2 exhibited less catalytic activity compared with that of the WT, and mutations in VR3 did not affect the activity, suggesting a less important role of VR3 of LotA in recognition of K6-linked diubiquitin chains. Similar to our findings, most amino acids that render the ubiquitin binding are reported from VR1 of Lots (Fig S6C–E). Therefore, our results indicate that the helical loop present toward the C-terminal from the CR is the VR1 of the Lot family, and it forms a part of the S1 ubiquitin-binding site. It is worth noting that similar findings on LotA are reported, whereas our article was under review (Warren et al, 2022 Preprint). Consistent with our finding, the crystal structure of LotA in complex with ubiquitin showed the VR1 of LotA to be crucial for ubiquitin binding to LotA.

Next, we investigated the effect of ubiquitin-binding sites of the OTU1 domain in the cooperative cleavage of K48-linked tetraubiquitin chains by OTU2 (Fig 7D and E). We introduced mutations in residues that we identified as ubiquitin-interacting residues on VR1 of the OTU1 domain into the LotA OTU1_OTU2 construct (LotA$_{7-544}$). Surprisingly, some of these mutants exhibited different processing speeds for K48-linked tetraubiquitin cleavage, supporting the cooperation between the OTU1 and OTU2 domains. For further characterization, we deleted the VR1 loop of OTU1 from the OTU1_OTU2 construct and examined its catalytic activity (Fig 7F). The VR1 deletion mutant exhibited faster cleavage of the K48-linked chains than the WT, indicating the inhibitory roles of VR1 of OTU1 in K48-linked chain cleavage by OTU2. Overall, our results strongly support that the OTU1 and OTU2 domains of LotA simultaneously interact with each other to regulate the cleavage of K48-linked polyubiquitin chains.

## Discussion

In this study, we investigated the structural and molecular basis of Lot DUBs. Structural analysis, including elucidation of the LotA OTU1 crystal structure, revealed that the EHL is found only in the Lot family (except LotA OTU1), shares the same structural topology, and provides the VR1, which acts as a ubiquitin-binding region. We also

---

presented as the mean ± S.D (n = 3, * 0.01 < P < 0.05 and ** 0.001 < P < 0.01). **(F)** Gel-based K48-linked polyubiquitin cleavage assay by LotA OTU1_OTU2 WT and VR1 deletion of LotA (residues 125–131) mutant of LotA OTU1_OTU2.
Source data are available for this figure.

elucidated that the catalytic triad of the Lot family (except LotA OTU1) has a specific arrangement similar to that of A20-type OTU-DUBs. These unique molecular and structural characteristics categorize Lot DUBs as a unique OTU subfamily. Moreover, the activity of both LotA OTU1 and OTU2 domains is specific for certain ubiquitin chains, and LotA OTU1 assists LotA OTU2 in catalyzing the cleavage of K48- or K63-linked polyubiquitin chains. Thus, the structural and biochemical analyses presented in this study provide the conserved molecular basis of the Lot DUB family and unravel another aspect of the regulatory mechanism of DUBs.

DUBs, with linkage specificity, cleave specific polyubiquitin linkages. The ability to distinguish between different polyubiquitin linkages depends on various molecular determinants (Faesen et al, 2011; Mevissen et al, 2013; Abdul Rehman et al, 2016; Gersch et al, 2017; Kwasna et al, 2018). Our study revealed an additional non-canonical regulatory mechanism in DUBs. Unlike other DUBs that have additional ubiquitin-binding sites near the catalytic center or within the catalytic domain, LotA exhibits intramolecular cooperativeness between its two OTU domains. The OTU1 domain specifically cleaves K6-linked polyubiquitin chains, but it is also critical for the cleavage of K48-linked polyubiquitin chains by LotA OTU2. Moreover, structural restriction induced by the deletion of the linker between OTU1 and OTU2 resulted in the reduced activity of LotA OTU2, indicating the cooperativeness between the two OTU domains of LotA.

Cooperative processing of K48-linked polyubiquitin chains by LotA is dependent on the length of the ubiquitin chain (Fig 2). We observed that intact LotA OTU1_OTU2 could not process K48-linked diubiquitin chains, but it could cleave K48-linked tetraubiquitin chains, indicating a length-dependent regulatory mechanism. Interestingly, recent studies have shown preferential cleavage of long K48-linked ubiquitin chains by the human DUBs MINDY1 and MINDY2 (Abdul Rehman et al, 2016; Abdul Rehman et al, 2021). MINDY1 has five different ubiquitin-binding sites (S1 and S1′–S4′) that act as the determinants of the length of the K48-linked polyubiquitin chain. Similar to that of MINDY1, we observed that the S1 ubiquitin-binding site of the OTU1 domain affects the activity of the OTU2 domain. We observed increased cleavage of K48-linked tetraubiquitin chains upon introduction of mutations in the VR1 of the OTU1 domain, which was identified as a K6-linked ubiquitin-binding site, suggesting an additional role of VR1 in the regulation of the cleavage of K48-linked polyubiquitin chains. Unlike MINDY1/2, which has five ubiquitin-binding sites, LotA has two catalytic domains and the ubiquitin-binding site on one catalytic domain regulates the other OTU domain. To further understand the intramolecular cooperation of LotA, structural analyses of K48-linked tetraubiquitin or longer chains with LotA OTU1_OTU2 are awaited.

Accumulating evidence supports that *L. pneumophila* harbors a series of genes that alter the host ubiquitination system. Among them, many DUBs that cleave specific polyubiquitin chains have been identified. *Legionella* has an arsenal of DUBs that interrupt all possible polyubiquitin linkages (LotA [K6, K48], LotB [K63], LotC [K6, K11, K27, K29, K33, K48, K63], LotD [K6, K11, K48, K63], and RavD [M1]) (Kubori et al, 2018; Wan et al, 2019; Shin et al, 2020a; Liu et al, 2020; Ma et al, 2020; Schubert et al, 2020; Takekawa et al, 2022). Although previous studies have revealed the cellular localization or interacting host proteins of these DUBs, little is known about their

physiological or pathological roles in infection. Therefore, systematic approaches are required to understand the underlying mechanism of DUBs in altering the host ubiquitination system.

# Materials and Methods

### Protein expression and purification

All proteins used in this study were expressed and purified as previously reported (Bhogaraju et al, 2016; Qiu et al, 2016). Lpg2248 (LotA), Lpg1621 (LotB), and Lpg2529 (LotC) were cloned into either a pParallelHis2 or pParallelGST2 vector (Sheffield et al, 1999). BL21(DE3) *Escherichia coli*–competent cells (NEB) were transformed with Lpg2248 (LotA) plasmids, and T7 Express *E. coli*–competent cells (NEB) were transformed with Lpg1621 (LotB) and Lpg2529 (LotC) plasmids, and both were grown in the LB medium to an optical density (600 nm) of 0.6–0.8 at 37°C. After reaching an optical density of 0.6–0.8, protein expression was induced by the addition of 0.5 mM IPTG (isopropyl $\beta$-D-1-thiogalactopyranoside) for an additional 16 h at 18°C and harvested. The cell pellet was resuspended in 50 mM Tris–HCl (pH 7.6), 150 mM NaCl, and 2 mM DTT (lysis buffer), and lysed by sonication and centrifuged at 12,000$g$ to defecate the supernatant. The supernatant of the GST-tagged protein was incubated for 2 h with glutathione Sepharose 4B (Cytiva), and pre-equilibrated with 50 mM Tris–HCl (pH 7.6), 500 mM NaCl, and 2 mM DTT (wash buffer); non-specific proteins were cleared by washing steps. Proteins were eluted with 50 mM Tris–HCl (pH 8.0), 50 mM NaCl, 2 mM DTT, and 15 mM L-glutathione reduced (elution buffer), and the buffer was exchanged to 50 mM Tris–HCl (pH 7.6), 150 mM NaCl, and 1 mM DTT (storage buffer). For His-tagged proteins, the supernatant was incubated for 2 h with TALON Metal Affinity Resin (Takara) pre-equilibrated with 50 mM Tris–HCl (pH 7.6), 500 mM NaCl, 10 mM imidazole, and 1 mM TCEP (wash buffer). His-tagged proteins were eluted with 50 mM Tris–HCl (pH 7.6), 500 mM NaCl, 300 mM imidazole, and 1 mM TCEP (elution buffer), and the buffer was exchanged to the storage buffer. For LotA$_{7–290}$, glutathione beads were incubated with sfGFP-TEV protease (Wu et al, 2009) for 4 h at 25°C and then overnight at 4°C, instead of using the elution buffer. GST-tagged cleaved proteins were exchanged to a buffer containing 20 mM Tris–HCl (pH 8.0), 20 mM NaCl, and 1 mM DTT (IEX buffer A) and purified by anion-exchange chromatography on HiTrap Q (Cytiva) with gradient elution using 20 mM Tris–HCl (pH 8.0), 1 M NaCl, and 1 mM DTT (IEX buffer B). Selected proteins were loaded onto a size-exclusion column (Superdex 75 16/60; Cytiva), and pre-equilibrated with 50 mM Tris–HCl (pH 7.6), 50 mM NaCl, and 1 mM TCEP (SEC buffer). For crystallization, proteins were concentrated at 267 $\mu$M and stored.

### Crystallization

Purified and concentrated LotA$_{7–290}$ were screened with the sitting-drop/vapor diffusion method, and screened in a 96-well plate with 100 nl of protein and 200 nl of precipitant solution at 290 K. Initial crystals were found from solution containing 32% (wt/vol) PEG 4000, 0.1 M Tris–HCl (pH 8.5), and 0.8 M lithium chloride with 267 $\mu$M

protein concentration. Diffraction-quality crystals were obtained from the optimized solution containing 35% (wt/vol) PEG 4000, 0.1 M Tris–HCl (pH 7.8), and 0.8 M lithium chloride with 267 $\mu$M protein concentration.

### Data collection and structure determination

Crystals were cryo-protected using a mother liquor solution containing 25% (vol/vol) ethylene glycol. Diffraction data were collected at the Pohang Light Source Beamline 11C. Initial datasets were processed using molecular replacement (McCoy et al, 2007) with Phaser in CCP4, using the prediction structure of AlphaFold2 Colab (Mirdita et al, 2022). Structure refinement and manual model building were performed with Coot and Phenix. Refine (Emsley et al, 2010; Afonine et al, 2012). All figures in this study were generated using PyMOL.

### Di-Ub panel cleavage assay

All diubiquitin chains used in this study were synthesized as previously described (El Oualid et al, 2010). To activate His-tagged LotA$_{7-290}$ and LotA$_{7-290}$ mutants, 3 $\mu$l of 1 $\mu$M DUBs was mixed with 12 $\mu$l of 25 mM Tris–HCl (pH 7.5), 150 mM NaCl, and 10 mM DTT (activation buffer) and incubated for 15 min at 25°C. For His-tagged LotA$_{7-544}$, LotA$_{294-544}$, LotB, and LotC, 3 $\mu$l of 5 $\mu$M DUBs was mixed with 12 $\mu$l of activation buffer. For diubiquitin samples, 3 $\mu$l of diubiquitin chains (0.2 mg/ml) was mixed with 3 $\mu$l of 500 mM Tris–HCl (pH 7.5), 500 mM NaCl, and 50 mM DTT (reaction buffer), and 12 $\mu$l of ultra-pure water. Reactions were initiated by mixed activated DUBs with diubiquitin and incubated at 37°C, and then, samples were collected at the indicated time-points. The reactions were quenched by the addition of 2X SDS sample buffer, then analyzed by SDS–PAGE, and stained with a silver staining kit (Power Silver Stain Kit; ELPIS Biotech).

### Poly-Ub panel cleavage assay

K48, K63-linked polyubiquitin chains used in this study were purchased from South Bay Bio. To activate Gst- or His-tagged LotA$_{7-290}$, LotA$_{7-544}$, and LotA$_{294-544}$, 3 $\mu$l of 5 $\mu$M DUBs was mixed with 12 $\mu$l of 25 mM Tris–HCl (pH 7.5), 150 mM NaCl, and 10 mM DTT (activation buffer) and incubated for 15 min at 25°C. For polyubiquitin samples, 3 $\mu$l of diubiquitin chains (0.2 mg/ml) was mixed with 3 $\mu$l of 500 mM Tris–HCl (pH 7.5), 500 mM NaCl, and 50 mM DTT (reaction buffer), and 12 $\mu$l of ultra-pure water. Reactions were initiated by mixed activated DUBs with polyubiquitin and incubated at 37°C, and then, samples were collected at the indicated time-points. The reactions were quenched by the addition of 2X SDS sample buffer, then analyzed by SDS–PAGE, and stained with a silver staining kit (Power Silver Stain Kit; ELPIS Biotech).

### (di)Ub-Prg, Ub-VA reactivity assay

K48, K63-linked diubiquitin Prg, and K48, K63-linked diubiquitin VA used in this study were synthesized as previously described (Mulder et al, 2014; Flierman et al, 2016). To activate His-tagged LotA$_{7-290}$, LotA$_{294-544}$, and LotA$_{7-544}$, 3 $\mu$l of 5 $\mu$M DUBs was mixed with

12 $\mu$l of 25 mM Tris–HCl (pH 7.5), 150 mM NaCl, and 10 mM DTT (activation buffer) and incubated for 15 min at 25°C. For diubiquitin Prg, VA probes, 3 $\mu$l of diubiquitin probe (0.5 mg/ml) was mixed with 3 $\mu$l of 500 mM Tris–HCl (pH 7.5), 500 mM NaCl, and 50 mM DTT (reaction buffer), and 12 $\mu$l of ultra-pure water. Reactions were initiated by mixed activated DUBs with a diubiquitin probe and incubated at 37°C, and then, samples were collected at the indicated time-points. The reactions were quenched by the addition of 2X SDS sample buffer, then analyzed by SDS–PAGE, and stained with a silver staining kit (Power Silver Stain Kit; ELPIS Biotech).

### Negative-stain electron microscopy

LotA$_{7-544}$ was diluted to 0.03 mg/ml in SEC buffer for negative staining. Four microliters of protein was loaded onto a glow-discharged CF200-CU carbon grid (Electron Microscopy Sciences). After 5 min of incubation, grid-bound LotA7–544 was quickly washed with distilled water. To stain Lot$_{7-544}$ molecules, 1% uranyl acetate was loaded onto the grid and incubated for 2 min. The stained grid was imaged using a JEOL-2100 Plus 200 kV microscope (JEOL) equipped with a Rio9 detector (Gatan). 60 micrographs were collected and processed using CryoSAPRC (Punjani et al, 2017). The extracted initial 7,535 particles were used for two-dimensional (2D) classification. The selected 1,594 particles of the 2D class are described in Fig 3A.

### Specimen preparation and cryo-EM data collection

The cryo-EM grid sample was prepared by diluting LotA$_{7-544}$ from 2 to 0.6 mg/ml in SEC buffer for cryo-EM. Quantifoil Au 1.2/1.3 200 mesh grids (Quantifoil) were glow-discharged for 60 s using a Pelco easiGlow Glow Discharge Cleaning System (Ted Pella, Inc.). Four microliters of LotA$_{7-544}$ was loaded onto a glow-discharged grid and blotted for 3 s with blot force 3 using Vitrobot Mark IV (Thermo Fisher Scientific).

C-clipped LotA$_{7-544}$ cryo-grid samples were imaged using a Glacios transmission electron microscope (Thermo Fisher Scientific) equipped with a field-emission gun with a voltage of 200 kV and a Falcon IV detector (Thermo Fisher Scientific). LotA$_{7-544}$ micrographs were obtained using a Falcon IV detector and automated data collection system EPU software (Thermo Fisher Scientific). LotA$_{7-544}$ was observed at a magnification of 150,000× with a pixel size of 0.68 Å. Each micrograph was recorded for 5.68 s, and the total dose was 60.14 e$^-$/Å$^2$ (~1 e$^-$/Å$^2$ per frame); the dose rate was 10.59 e$^-$/Å$^2$/s. Lot$_{7-544}$ was imaged with a defocus range from −1.0 to −1.8 $\mu$m and total fractionation as 60 frames per micrograph.

### Image processing

Computing resources were used in CMCI at SNU and GSDC at KISTI. For LotA7–544, 2,538 micrographs were obtained EPU software. Initial dataset processing was performed using CryoSPARC software (Punjani et al, 2017). This dataset was motion-corrected to 6–55 movie frames per micrograph. Motion-corrected micrographs were used to estimate the contrast transfer function (CTF) using CTFFIND4 (Rohou & Grigorieff, 2015). The CTF-estimated micrographs were filtered into the final 1,796 micrographs for data

processing. We used 400 micrographs for full-set particle picking. Next, we performed deep learning–based particle-picking software (Topaz train) (Bepler et al, 2020) using reference particles. A total of 497,860 particles were used for the two-round 2D classification. After 2D classification, the selected 176,036 particles were transferred from *CryoSPARC* to *RELION 3.1.3* (Scheres, 2012; Zivanov et al, 2018) and further processed. The 2D classification was used for the selected particles. The final particles (113,381 particles) were used for an initial three-dimensional (3D) model and classification; 3D classification was classified into three classes, and the model was refined. This model was deposited in the EMDB (EMDB: 34350).

## Data Availability

Structural information including atomic coordinates and structure factors for the LotA OTU1 (LotA$_{7–290}$) structure is deposited in the Protein Data Bank (http://www.rcsb.org/) under an accession number 8GOK.

## Supplementary Information

## Acknowledgements

We thank Sagar Bhogaraju and Sissy Kalayil for the critical reading of the article. We also thank Prof. Hyun Soo Cho for sharing synchrotron time. The authors also thank the staff at PAL 5C and 11C for their support during the crystallographic X-ray diffraction test and data collection. This work was supported by a National Research Foundation of Korea (NRF) grant funded by the Korean government (MSIT and MOE) (Nos. 2021R1C1C1003961, 2018R1A6A1A03025607, and 2021M3A9I4021220, 2021R1I1A1A01049284, and 2021R1A6C103B381) and the Yonsei University Research Fund of 2021 (2021-22-0050). GJ van der Heden van Noort acknowledges funding by ZonMw (Off-road grant 451001026) and NWO (VIDI grant VI.VIDI.192.011).

### Author Contributions

S Kang: conceptualization, data curation, investigation, and writing—original draft, review, and editing.
G Kim: data curation and investigation.
M Choi: investigation.
M Jeong: investigation.
GJ van der Heden van Noort: supervision, funding acquisition, investigation, and writing—review and editing.
S-H Roh: supervision, funding acquisition, investigation, and writing—review and editing.
D Shin: conceptualization, supervision, funding acquisition, investigation, project administration, and writing—original draft, review, and editing.

## Conflict of Interest Statement

The authors declare that they have no conflict of interest.

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
