## [Reviewer comments · Life Science Alliance]

Life Science Alliance

Structural insights into ubiquitin chain cleavage by *Legionella* ovarian tumor deubiquitinases

Sangwoo Kang, Gyuhee Kim, Minhyung Choi, Minwoo Jeong, Gerbrand van der Heden van Noort, Soung-Hun roh, and Donghyuk Shin

DOI: <https://doi.org/10.26508/lsa.202201876>

Corresponding author(s): Donghyuk Shin, Yonsei University

Review Timeline:

Submission Date:	2022-12-15
Editorial Decision:	2023-01-06
Revision Received:	2023-03-23
Editorial Decision:	2023-04-11
Revision Received:	2023-04-17
Accepted:	2023-04-17

Transaction Report:

January 6, 2023

Re: Life Science Alliance manuscript #LSA-2022-01876-T

Donghyuk Shin
Yonsei University
Department of Systems Biology
Yonsei-ro 50
Seoul, Seoul 03722
Korea, Republic of

Dear Dr. Shin,

Thank you for submitting your manuscript entitled "Structural insights into ubiquitin chain cleavage by Legionella ovarian tumor deubiquitinases" to Life Science Alliance. The manuscript was assessed by expert reviewers, whose comments are appended to this letter. We invite you to submit a revised manuscript addressing the Reviewer comments.

Thank you for this interesting contribution to Life Science Alliance. We are looking forward to receiving your revised manuscript.

Sincerely,

B. MANUSCRIPT ORGANIZATION AND FORMATTING:

Reviewer #1 (Comments to the Authors (Required)):

In this study, the authors aim to shed some light on the structural and molecular basis of ubiquitin chain cleavage mediated by Legionella Lot deubiquitinases. The latter (Lot DUBs) belong to a family containing four members (Lots A, B, C and D), that are similar to human OTU (ovarian tumor domain containing) type deubiquitinases, also possessing OTU domains. The authors use multiple approaches to obtain structural information: they determine the cryo-EM structure of LotA-OTU1-OTU2, as well as the crystal structure of LotA-OTU1. In doing so, and by making comparisons with other Lot enzymes, they conclude that Lot DUBs differ from OTU DUBs both with respect to their overall structural topology and also with respect to their catalytic triads (for example, they conclude that the EHL is found only in Lot DUBs, which also provides the VR1 binding site for ubiquitin). Throughout the study, the authors also use biochemical approaches to perform ubiquitin cleavage assays.

Overall, the authors manage to describe some of the distinct structural characteristics of Lot DUBs and shed new light on catalysis. Their findings are novel and interesting, but in my opinion, further work (or at least more discussion) is required to explain the structural/molecular basis of the enzymes' preference (selectivity) for different types of ubiquitin chains.

Also, I have encountered mislabeling in several instances along the manuscript/figures.

Figures 1 and 2: The authors first perform di-ubiquitin cleavage assays using purified proteins. They also use K48- or K63-linked di-ubiquitin activity-based probes. In line with the literature, they confirm that LotA-OTU1 domain cleaves K6-linked di-ubiquitin. However, not supporting the literature, they show that neither LotA-OTU2 nor LotA-OTU1-OTU2 can cleave K48- or K63-linked di-ubiquitin. Thus, they employ tetra-ubiquitin chains as substrates.

Comments to the authors:

- Fig 2A: In this K48-linked di-ubiquitin cleavage assay, it is quite confusing that the dipeptide levels are decreasing upon incubation with LotA-OTU2, and even more so with LotA-OTU1-OTU2. This is in contrast with what the authors show earlier in Fig 1C and Fig 1D. What is the authors' explanation for this?
- Figs 2B and 2C: LotA-OTU2 alone does not seem to cleave K48- or K63-linked tetra-Ub chains (contrary to what the authors claim in line 130). Looking at the figure, this cleavage activity seems to be very weak, if non-existent. The authors should tone down their sentence (line 130), as they actually did in the table of Fig 2E. Plus, even when OTU1 is together with OTU2, the cleavage activity against K63-linked tetra-Ub is still very weak (yet, the authors chose to call this weak activity "moderate" in the table in Fig 2E). I think that these conclusions should be toned down.
- Fig 2E (table): Some of the conclusions in the current table are not in line with what the results indicate. As far as OTU2 is concerned, I don't think that the authors have any evidence which indicates that this domain has any activity, weak or not, against tri-ubiquitin chains. Instead of "Ub3 or longer", they should simply say "Ub4 or longer". Looking at the last panel of Fig 2B, only when OTU1 and OTU2 are together, there is a decrease in the level of K48-Ub3. However, the same cannot be said for K63-Ub3. Thus, the authors should update the table accordingly. For instance, for OTU1-OTU2, obviously they should not say "Ub2 or longer" (for K48 and K63, because they have already claimed in Fig 1 that OTU1-OTU2 has no activity against the dipeptide). They should say "Ub3 or longer" for K48, and "Ub4 or longer" for K63.

Figure 3: The authors perform Cryo-EM on LotA-OTU1-OTU2. Based on their images and a recently announced crystal structure, which suggests 2 different conformations for the OTU2 domain when the OTU1 structures are superimposed, they deleted a stretch of aminoacids (276-283) to restrict the movement of the two domains (OTU1 and OTU2) with respect to each other.

Comments to the authors:

- Fig 3D: Why do the figure and the figure legends call this deletion mutant DEL276-279 while the text says DEL276-283? Which is true?
- Fig 3D: The authors claim that DEL276-279 (or is it DEL276-283?) cleaves K48-linked tetra-ubiquitin chains more slowly. This

is not obvious, looking at the gel. This is an important claim and requires a more representative image. In fact, a graph that represents quantifications from several experiments, comparing WT LotA-OTU1-OTU2 and its deletion mutant, along with error bars, would be more appropriate. Did the authors perform this experiment also with K68-linked tetra-ubiquitin? Because the activity of WT LotA-OTU1-OTU2 against this peptide is already very weak (again, not moderate), it would be interesting to compare it with the linker deletion mutant, which may now display total activity loss.

Figure 4: The authors perform X-ray crystallography, along with molecular replacement using AlphaFold, to determine the crystal structure of Lot1-OTU1 at 1.54 Å resolution. Both an OTU fold and an EHL (91-199) are recognizable in their structure. They then perform structural similarity assays (SSA) between Lot enzymes with known structures (LotA-OTU2, LotB, LotC), and conclude that they all possess an EHL, and that their overall topologies are similar.

Comments to the authors: I agree with the authors that the EHL seems to be common to Lot proteases. However, the authors go a step further and claim that EHL is a unique structural feature that distinguishes Lot proteases from the other DUBs. In order to support their claims, they should include a few "non-Lot" DUBs in their structural similarity assays for comparison. This should be a rather straightforward and easy analysis to perform.

Figure 7: In this figure, the authors further characterize the S1 Ubiquitin binding site in the OTU fold of LotA-OTU1 domain (based on their crystal structure). Because all the Lot enzymes have EHL, and also because the EHL is close to S1, they investigated the effect of EHL itself on S1. They identified a helical loop in the C-end of the EHL core helix (which seems to be conserved) and the authors believe that this may actually be the VR1 region. Here, they mutated several residues and showed that 8 of these mutations (out of 12) resulted in reduced K6-linked di-ubiquitin cleavage activity. In addition, 2 mutations in VR2 also impaired the cleavage activity against this peptide.

Comments to the authors:

- How do the authors explain that none of the mutations in VR3 affects K6-Ub2 cleavage? Is VR3 irrelevant for Ub binding to LotA-OTU1?
- Fig 7C: I believe that the lower gel images are mislabeled. They should be LotA-OTU1 VR2 and VR3 mutants, not LotA-OTU2 !

Final comment: As mentioned earlier, this manuscript provides novel and important insight into the structural basis of ubiquitin chain cleavage by Lot proteases, and should be published provided that the authors address the above points. Nevertheless, the authors still fall short of putting their findings in a larger context and do not really explain or discuss in detail the basis of specificity and selectivity for Lot enzymes (or their OTU domains) as far as different ubiquitin linkages and chain lengths are concerned.

For example, the crystal structures of K68-linked di-ubiquitin and tri-ubiquitin chains are available. Similarly, the structures of K48-linked di-ubiquitin and tetra-ubiquitin chains are also available. Would it be possible to perform some docking analyses using these Ubiquitin structures and the structures of different Lot enzymes (or at least, the authors' own structure) so that we can gain further insight into the basis of chain length and type selectivity?

'Referee Cross-Comments'

In my opinion, the points raised by both reviewers are fair and justified, and they are mostly in line with my own comments. My only minor comment would be in response to Reviewer 3's statement "There is no attempt to combine with Legionella infection assays"

I agree with reviewer 3 in that these experiments would be of great value, but I also doubt that the authors would have the necessary set up and expertise to do it.

Reviewer #2 (Comments to the Authors (Required)):

In this study, Kang et al. examine the Legionella Lot proteases, an OTU deubiquitinating enzyme (DUB) subfamily that display various ubiquitin (Ub) chain linkage specificities, although little is known about their function as effectors during bacterial infection of human or other host cells. LotA is an unusual member of this group of enzymes in that it has two OTU domains, OTU1 and OTU2. It was known that OTU1 cleaves K6-linked Ub chains, and OTU2 or OTU1-OTU2 had been reported to cleave K48 and K63 chains. However, the current authors tested K48 and K63 diubiquitin chains and saw no cleavage, but when they tested longer chains with these linkages, cleavage was observed. Most interestingly, even though OTU1 catalytic activity was not needed, OTU1-OTU2 cleaved these longer chains much more efficiently than did OTU2 alone.

A crystal structure of LotA OTU1-OTU2 was deposited in the PDB by another group last year; comparison with a low resolution cryoEM structure in the current work suggest the two domains can assume various positions relative to one another. A high-resolution OTU1 crystal structure was also determined in the current work and was compared to other Lot OTU domain structures. An insert called the EHL domain was noted to be common to this group and proposed to include a Ub-binding VR1. Mutations in VR1 impaired OTU1 cleavage of K6 diUb but surprisingly, appeared to enhance cleavage of K48 Ub4 chains by OTU1-OTU2. This supports the idea that OTU1 cooperates with OTU2 to promote cleavage of these longer chains, although without solving structures of OTU1-OTU2 bound to Ub or Ub chains, the mechanism will remain unclear. Also, as noted below, some of these proposed cleavage differences were hard to discern in the gels shown.

This manuscript provides both useful structural and biochemical insights into the Legionella Lot OTU enzymes. The general quality of experiments is high except that in a few places, enzymatic changes are inferred for certain mutants where more quantitative analysis will be needed to solidify the claims. A manuscript was just published in *Molecular Cell* by Pruneda and colleagues on LotA; a direct comparison of results would be important to include in the revised manuscript (there are some significant differences, such as the inference of the third residue of the catalytic triad). Overall, with some modest revisions (particularly, quantitation in a few critical places), I am in favor of publication.

Specific comments:

-line 130 Activity of OTU2 toward K63 Ub4 is not visible in the Fig. 2C that I have.

-Fig. 2E summary says OTU1-OTU2 activity is strong with Ub2 and longer but NO activity seen with Ub2 in Fig. 1. Maybe I'm misreading the Fig. 2E table.

-Why are the Ub trimers spread into multiple bands in Fig. 2D, 3D?

-line 115 "form a covalently linked complex" - what does this mean? The figure panels refer to describe diUb cleavage assays.

-Figures 3 and 4 are out of order.

-Fig. 3D: The slower kinetics for the linker deletion (between the OTU domains) are not obvious. These blots therefore need quantification.

-Similarly, in Fig. 7D, E, I don't see any substantial differences from WT. This HAS to be done quantitatively and with suitable statistical analysis of repeats.

-line 376: Preferential cleavage of longer Ub chains is also seen in a bacterial CE clan protease OtDUB (Berk et al. *Nature Comm* 2020). Interestingly, the Warren et al. (2022) *Mol Cell* paper notes that the UBD/EHL in LotA is similar topologically to a UBD in OtDUB.

Reviewer #3 (Comments to the Authors (Required)):

This is a biochemical and structural paper concerning the DUB activity encoded by a Legionella gene LotA. Some aspects overlap with a recent paper from the Pruneda laboratory published in *Molecular Cell*. This DUB is unusual in having two catalytic sites belonging to the OTU family, here labelled OTU1 and OTU2. OTU1 highly stringent for Lys6 ubiquitin chain linkages, whilst OTU2 processes other linkage types and requires a longer chain length (known in 2018, Kubori et al.). The OTU1 specificity provides a useful analytical tool and this has been comprehensively described in the Pruneda paper. The majority of reported results are confirmatory. The main novelty contained in this paper is the involvement of OTU1 in augmenting the catalytic activity of OTU2 in a non-catalytic manner (shown by mutation of OTU1 catalytic cysteine). By and large the experiments are carefully done and will be of relatively narrow specialist interest to the DUB field. There is no attempt to combine with Legionella infection assays, nor to obtain structures with ubiquitin bound (could this have been an informative element of their cryo-EM studies?).

Major comments:

the failure of both catalytic domains to bind an active site probe contradicts the Pruneda paper. This issue should be resolved or commented upon.

In figure 1C the processing of K6 by OTU2 is pretty much as good as the double domain protein 1D. Why is that and why do we not see the emergence of monoUb to the same extent in Fig1 as we do in later figures (e.g. 5E)?

Figure 2A the loading is off.

Minor comments:

line 46-47, could perhaps be more careful in these specificities, for example K63 chains are clearly involved in DNA repair and evidence for a role of K6 chains in mitophagy is relatively weak. Line 113-115 - something wrong here.

For the non specialist S1, S1" and S2 sites should be defined.

Reviewer #1

In this study, the authors aim to shed some light on the structural and molecular basis of ubiquitin chain cleavage mediated by Legionella Lot deubiquitinases. The latter (Lot DUBs) belong to a family containing four members (Lots A, B, C and D), that are similar to human OTU (ovarian tumor domain containing) type deubiquitinases, also possessing OTU domains. The authors use multiple approaches to obtain structural information: they determine the cryo-EM structure of LotA-OTU1-OTU2, as well as the crystal structure of LotA-OTU1. In doing so, and by making comparisons with other Lot enzymes, they conclude that Lot DUBs differ from OTU DUBs both with respect to their overall structural topology and also with respect to their catalytic triads (for example, they conclude that the EHL is found only in Lot DUBs, which also provides the VR1 binding site for ubiquitin). Throughout the study, the authors also use biochemical approaches to perform ubiquitin cleavage assays.

Overall, the authors manage to describe some of the distinct structural characteristics of Lot DUBs and shed new light on catalysis. Their findings are novel and interesting, but in my opinion, further work (or at least more discussion) is required to explain the structural/molecular basis of the enzymes' preference (selectivity) for different types of ubiquitin chains.

Also, I have encountered mislabeling in several instances along the manuscript/figures.

Figures 1 and 2: The authors first perform di-ubiquitin cleavage assays using purified proteins. They also use K48- or K63-linked di-ubiquitin activity-based probes. In line with the literature, they confirm that LotA-OTU1 domain cleaves K6-linked di-ubiquitin. However, not supporting the literature, they show that neither LotA-OTU2 nor LotA-OTU1-OTU2 can cleave K48- or K63-linked di-ubiquitin. Thus, they employ tetra-ubiquitin chains as substrates.

→ Overall, we highly appreciated specific and valuable comments from the reviewer. Suggested comments significantly improved our manuscript.

Comments to the authors:

- Fig 2A: In this K48-linked di-ubiquitin cleavage assay, it is quite confusing that the dipeptide levels are decreasing upon incubation with LotA-OTU2, and even more so with LotA-OTU1-OTU2. This is in contrast with what the authors show earlier in Fig 1C and Fig 1D. What is the authors' explanation for this?

→ We thank the reviewer for this specific comment. We have repeated the di-ubiquitin cleavage assay and confirmed that both LotA-OTU2 and LotA-OTU1-OTU2 do not cleave K48-linked di-ubiquitin. (New Figures: Fig. 1B, 1C, 1D, Fig. 2A)

- Figs 2B and 2C: LotA-OTU2 alone does not seem to cleave K48- or K63-linked tetra-Ub chains (contrary to what the authors claim in line 130). Looking at the figure, this cleavage activity seems to be very weak, if non-existent. The authors should tone down their sentence (line 130), as they actually did in the table of Fig 2E. Plus, even when OTU1 is together with OTU2, the cleavage activity against K63-linked tetra-Ub is still very weak (yet, the authors chose to call this weak activity "moderate" in the table in Fig 2E). I think that these conclusions should be toned down.

→ We thank the reviewer for this specific comment. We agree that the intrinsic activity of OTU2 against both K48- and K63- linked tetra ubiquitin chains. The activity of OTU1_OTU2 against K63-linked tetra-Ub is also very weak. We repeated the experiment and these results were reproducible. We have toned down and changed the wording in the Fig 2E. table (New Figures: Fig. 2E).

- Fig 2E (table): Some of the conclusions in the current table are not in line with what the results indicate. As far as OTU2 is concerned, I don't think that the authors have any evidence which indicates that this domain has any activity, weak or not, against tri-ubiquitin chains. Instead of "Ub3 or longer", they should simply say "Ub4 or longer". Looking at the last panel of Fig 2B, only when OTU1 and OTU2 are together, there is a decrease in the level of K48-Ub3. However, the same cannot be said for K63-Ub3. Thus, the authors should update the table accordingly. For instance, for OTU1-OTU2, obviously they should not say "Ub2 or longer" (for K48 and K63, because they have already claimed in Fig 1 that OTU1-OTU2 has no activity against the dipeptide). They should say "Ub3 or longer" for K48, and "Ub4 or longer" for K63.

→ We thank the reviewer for this specific comment. We agree with the reviewer and updated the wording in Fig. 2E (table).

Figure 3: The authors perform Cryo-EM on LotA-OTU1-OTU2. Based on their images and a recently announced crystal structure, which suggests 2 different conformations for the OTU2 domain when the OTU1 structures are superimposed, they deleted a stretch of amino acids (276-283) to restrict the movement of the two domains (OTU1 and OTU2) with respect to each other.

Comments to the authors:

- Fig 3D: Why do the figure and the figure legends call this deletion mutant DEL276-279 while the text says DEL276-283? Which is true?

→ We thank the reviewer for this specific comment. We have corrected the label. The mutant was DEL₂₇₆₋₂₈₃.

- Fig 3D: The authors claim that DEL276-279 (or is it DEL276-283?) cleaves K48-linked tetra-ubiquitin chains more slowly. This is not obvious, looking at the gel. This is an important claim and requires a more representative image. In fact, a graph that represents quantifications from several experiments, comparing WT LotA-OTU1-OTU2 and its deletion mutant, along with error bars, would be more appropriate. Did the authors perform this experiment also with K68-linked tetra-ubiquitin? Because the activity of WT LotA-OTU1-OTU2 against this peptide is already very weak (again, not moderate), it would be interesting to compare it with the linker deletion mutant, which may now display total activity loss.

→ We thank the reviewer for this critical and valuable comment. We have repeated the K48-linked tetra ubiquitin chain cleavage assay with deletion mutant. We quantified the gels and added the bar graph showing that the deletion mutant cleaves the K48-linked tetra ubiquitin more slowly than the wild-type. (New Figures: Fig. 3D, Fig. 3E) In addition, we also tested the K63-linked tetra-ubiquitin cleavage with the deletion mutant. However, as the reviewer pointed out, the catalytic activity of OTU1_OTU2 against the K63-tetra ubiquitin chain is too weak, and we could not find differences in catalytic activity between wild-type and deletion mutant (New Figures: Fig. EV2C, EV2D).

Figure 4: The authors perform X-ray crystallography, along with molecular replacement using AlphaFold, to determine the crystal structure of Lot1-OTU1 at 1.54 Å resolution. Both an OTU

fold and an EHL (91-199) are recognizable in their structure. They then perform structural similarity assays (SSA) between Lot enzymes with known structures (LotA-OTU2, LotB, LotC), and conclude that they all possess an EHL, and that their overall topologies are similar.

Comments to the authors: I agree with the authors that the EHL seems to be common to Lot proteases. However, the authors go a step further and claim that EHL is a unique structural feature that distinguishes Lot proteases from the other DUBs. In order to support their claims, they should include a few "non-Lot" DUBs in their structural similarity assays for comparison. This should be a rather straightforward and easy analysis to perform.

→ We thank the reviewer for this valuable comment. This comment helped to improve our manuscript. To examine whether the EHL domain is a unique structural motif for LOTs, we performed the structural similarity analysis of LotA-EHL against the entire protein structure database that included all the predicted structures (also all the known DUBs) from AlphaFold (FoldSeek). The analysis only picked LotA orthologues from other *Legionella* species (*L. antarctica* and *L. moravica*), but not in other species. It supports our finding that the EHL fold is unique to Lots. We have included this result in the revised manuscript (Line 215-219).

Figure 7: In this figure, the authors further characterize the S1 Ubiquitin binding site in the OTU fold of LotA-OTU1 domain (based on their crystal structure). Because all the Lot enzymes have EHL, and also because the EHL is close to S1, they investigated the effect of EHL itself on S1. They identified a helical loop in the C-end of the EHL core helix (which seems to be conserved) and the authors believe that this may actually be the VR1 region. Here, they mutated several residues and showed that 8 of these mutations (out of 12) resulted in reduced K6-linked di-ubiquitin cleavage activity. In addition, 2 mutations in VR2 also impaired the cleavage activity against this peptide.

Comments to the authors:

- How do the authors explain that none of the mutations in VR3 affects K6-Ub2 cleavage? Is VR3 irrelevant for Ub binding to LotA-OTU1?

→ We thank the reviewer for this specific comment. As the reviewer pointed out, the VR3 of LotA OTU1 seems to be not crucial for Ub binding. We have now mentioned this interpretation in the manuscript (Line 332-336).

- Fig 7C: I believe that the lower gel images are mislabeled. They should be LotA-OTU1 VR2 and VR3 mutants, not LotA-OTU2 !

→ Thanks to the reviewer for the correction. We have corrected the label.

Final comment: As mentioned earlier, this manuscript provides novel and important insight into the structural basis of ubiquitin chain cleavage by Lot proteases, and should be published provided that the authors address the above points. Nevertheless, the authors still fall short of putting their findings in a larger context and do not really explain or discuss in detail the basis of specificity and selectivity for Lot enzymes (or their OTU domains) as far as different ubiquitin linkages and chain lengths are concerned.

For example, the crystal structures of K68-linked di-ubiquitin and tri-ubiquitin chains are available. Similarly, the structures of K48-linked di-ubiquitin and tetra-ubiquitin chains are also available. Would it be possible to perform some docking analyses using these Ubiquitin

structures and the structures of different Lot enzymes (or at least, the authors' own structure) so that we can gain further insight into the basis of chain length and type selectivity?

→ We tried several docking analyses using Ubiquitin structures with LotA. However, as shown below (only for the reviewers), different docking programs gave us quite different poses of ubiquitin bindings. As the Ub-bound structure from Pruneda's laboratory is published during the review process, we cited the paper and compared their findings with our results (Line 338-341).

→

[Figure removed by LSA Editorial Staff per authors' request.]

'Referee Cross-Comments'

In my opinion, the points raised by both reviewers are fair and justified, and they are mostly in line with my own comments. My only minor comment would be in response to Reviewer 3's statement "There is no attempt to combine with Legionella infection assays"

I agree with reviewer 3 in that these experiments would be of great value, but I also doubt that the authors would have the necessary set up and expertise to do it.

→ We highly appreciated this specific comment from the reviewer. Because the biological roles of LotA have been reported in previous literature, our manuscript did not explore the roles of Lot-DUBs in infectious conditions. Our study aimed to reveal the fundamental basis of molecular details of Lot-DUBs.

Reviewer #2

In this study, Kang et al. examine the Legionella Lot proteases, an OTU deubiquitinating enzyme (DUB) subfamily that display various ubiquitin (Ub) chain linkage specificities, although little is known about their function as effectors during bacterial infection of human or other host cells. LotA is an unusual member of this group of enzymes in that it has two OTU domains, OTU1 and OTU2. It was known that OTU1 cleaves K6-linked Ub chains, and OTU2 or OTU1-OTU2 had been reported to cleave K48 and K63 chains. However, the current authors tested K48 and K63 diubiquitin chains and saw no cleavage, but when they tested longer chains with these linkages, cleavage was observed. Most interestingly, even though OTU1 catalytic activity was not needed, OTU1-OTU2 cleaved these longer chains much more efficiently than did OTU2 alone.

A crystal structure of LotA OTU1-OTU2 was deposited in the PDB by another group last year; comparison with a low resolution cryoEM structure in the current work suggest the two domains can assume various positions relative to one another. A high-resolution OTU1 crystal structure was also determined in the current work and was compared to other Lot OTU domain structures. An insert called the EHL domain was noted to be common to this group and proposed to include a Ub-binding VR1. Mutations in VR1 impaired OTU1 cleavage of K6 diUb but surprisingly, appeared to enhance cleavage of K48 Ub₄ chains by OTU1-OTU2. This supports the idea that OTU1 cooperates with OTU2 to promote cleavage of these longer chains, although without solving structures of OTU1-OTU2 bound to Ub or Ub chains, the mechanism will remain unclear. Also, as noted below, some these proposed cleavage differences were hard to discern in the gels shown.

This manuscript provides both useful structural and biochemical insights into the Legionella Lot OTU enzymes. The general quality of experiments is high except that in a few places, enzymatic changes are inferred for certain mutants where more quantitative analysis will be needed to solidify the claims. A manuscript was just published in Molecular Cell by Pruneda and colleagues on LotA; a direct comparison of results would be important to include in the revised manuscript (there are some significant differences, such as the inference of the third residue of the catalytic triad). Overall, with some modest revisions (particularly, quantitation in a few critical places), I am in favor of publication.

Specific comments:

-line 130 Activity of OTU2 toward K63 Ub₄ is not visible in the Fig. 2C that I have.

→ We agree with the reviewer. This point is also raised from the reviewer 1. We repeated the experiments for both K48- and K63- linked tetra Ub, and observed that the activity of LotA OTU2 against K63-linked tetra-Ub is only visible when the chain is incubated long enough. (Faint mono ubiquitin band from the O/N sample). Based on this observation and suggestion from reviewer 1, we also corrected the label in the Fig. 2E table.

-Fig. 2E summary says OTU1-OTU2 activity is strong with Ub₂ and longer but NO activity seen with Ub₂ in Fig. 1. Maybe I'm misreading the Fig. 2E table.

→ We thank the reviewer for this specific comment, we have corrected the table in Fig. 2E.

-Why are the Ub trimers spread into multiple bands in Fig. 2D, 3D?

→ We have repeated the experiments for Fig. 2D and 3D. The spreads of Ub trimers were always observed during our experiment. However, the multiple bands disappeared from di-Ubiquitin species. Similarly, K48-trimer smearing-bands were observed from previous literature (J.M. Berk et al, 2020 PMID: 32393759, Warren, et al, 2023, PMID: 36538933). Based on this observation, we assume that the branched conformation of K48-linked Ub3 may affect its mobility on the PAGE gel.

-line 115 "form a covalently linked complex" - what does this mean? The figure panels refer to describe diUb cleavage assays.

→ We thank the reviewer for this specific comment. We have corrected the sentence.

-Figures 3 and 4 are out of order.

→ We thank the reviewer for the specific comment; the order is corrected.

-Fig. 3D: The slower kinetics for the linker deletion (between the OTU domains) are not obvious. These blots therefore need quantification.

→ We thank the reviewer for this comment. We repeated the experiment and included the bar graph of quantified intensities with p-values (New Figures: Fig. 3D, Fig. 3E).

-Similarly, in Fig. 7D, E, I don't see any substantial differences from WT. This HAS to be done quantitatively and with suitable statistical analysis of repeats.

→ We thank the reviewer for the specific comment. We repeated the experiment and included the bar graph of quantified intensities with p-values (New Figures: Fig. 7D, 7E, 7F).

-line 376: Preferential cleavage of longer Ub chains is also seen in a bacterial CE clan protease OtDUB (Berk et al. Nature Comm 2020). Interestingly, the Warren et al. (2022) Mol Cell paper notes that the UBD/EHL in LotA is similar topologically to a UBD in OtDUB.

→ We thank the reviewer for this valuable comment. In the Mol Cell paper, they used dali server to find structurally similar proteins. In our revised manuscript, we performed the structural similarity analysis of LotA-EHL against the entire protein structure database that included all the predicted structures (also all the known DUBs) from Alphafold (FoldSeek). The analysis only picked LotA orthologues from other Legionella species (*L. antarctica* and *L. moravica*), but not in other species. It supports our finding that the EHL fold is unique to Lots. We have included this result in the revised manuscript (Line 215-219).

Of course, we fully agree with the finding from Pruneda's laboratory. They claimed that the OtDUB has a similar four helix bundle as a UBD domain. However, as they also mentioned in their manuscript the second ubiquitin-binding domain of OtDUB is not conserved in LotA OTU1 EHL and we think this is the reason why our structural analysis ruled out the OtDUB's UBD from similar folded structures to LotA EHL.

Reviewer #3

This is a biochemical and structural paper concerning the DUB activity encoded by a Legionella gene LotA. Some aspects overlap with a recent paper from the Pruneda laboratory published in Molecular Cell. This DUB is unusual in having two catalytic sites belonging to the OTU family, here labelled OTU1 and OTU2. OTU1 is highly stringent for Lys6 ubiquitin chain linkages, whilst OTU2 processes other linkage types and requires a longer chain length (known in 2018, Kubori et al.). The OTU1 specificity provides a useful analytical tool and this has been comprehensively described in the Pruneda paper. The majority of reported results are confirmatory. The main novelty contained in this paper is the involvement of OTU1 in augmenting the catalytic activity of OTU2 in a non-catalytic manner (shown by mutation of OTU1 catalytic cysteine). By and large the experiments are carefully done and will be of relatively narrow specialist interest to the DUB field. There is no attempt to combine with Legionella infection assays nor to obtain structures with ubiquitin bound (could this have been an informative element of their cryo-EM studies?).

→ We thank the reviewer for the specific and valuable comments. We agree with the reviewer's comments. As the reviewer pointed out, while our findings are supportive to Pruneda's story (Mol. Cell. 2023). However, our paper is submitted before their paper is published, and our story is complementary to and consistent with their findings and provides several points that support both stories.

We highly appreciated this specific comment from the reviewer. The biological roles of LotA have been reported in previous literature. The original LotA paper (Kubori 2018) showed that the deletion of LotA does not show a significant defect in *L. pneumophila* growth. Therefore, our manuscript did not explore the roles of Lot-DUBs in infectious conditions. Instead, our study aimed to reveal the fundamental basis of molecular details of Lot-DUBs.

We agree with the reviewer that the ubiquitin-bound structures would support our findings. Of course, before we submit our manuscript and even during the revision, we tried to get the cryo-EM images of LotA OTU1_OTU2 with K48-linked ubiquitin chains to visualize how OTU1 supports OTU2 in recognition of K48-linked Ub chains. However, for this complex structure, we had to have high-quality of pure K48-linked tri- or tetra-ubiquitin. It is because the LotA OTU1_OTU2 does not cleave K48-Ub2 and is not reactive to K48-Ub2-Prg or K48-VME-UB2 (EV Fig1 A). The preparation of K48-tri-Ub or K48-tetra-Ub was technically challenging as it should be done by enzymatic reactions or chemical synthesis. Though we tried several conditions, we could not get the complex structure of LotA OTU1_OTU2 with K48-Ub3 or K48-Ub4. We will continue to follow this story and we believe that the complex structure of LotA OTU1_OTU2:K48 Ub3(or Ub4) itself will become another full story as it will tell us the molecular details of how OTU1 and OTU2 interact with.

Major comments:

the failure of both catalytic domains to bind an active site probe contradicts the Pruneda paper. This issue should be resolved or commented upon.

→ We thank the reviewer for the specific comment. We have now included the new figure (Fig EV 1E) showing the OTU1, OTU2, and OTU1_OTU2 are reactive to Ub-Prg and form a covalent complex. This is consistent with Pruneda's paper. What we were showing in Fig EV 1 A-D was LotA constructs are not reactive to "di-ubiquitin" activity probes which were not examined in Pruneda's paper.

In figure 1C the processing of K6 by OTU2 is pretty much as good as the double domain protein 1D. Why is that and why do we not see the emergence of monoUb to the same extent in Fig1 as we do in later figures (e.g. 5E)?

→ We thank the reviewer for the specific comment. We repeated the experiments and we observed consistent results that also fit recent publication (Warren et al, 2023) LotA OTU1 specifically cleaves K6-linked Ub while LotA OTU2 itself hardly cleaves K6-Ub2. LotA OTU2 alone cleaves K48-UB4 and the presence of OTU1 enhances this cleavage.

Figure 2A the loading is off.

→ We thank the reviewer for the specific comment. We repeated the experiment and updated the figure with proper loading controls.

Minor comments:

line 46-47, could perhaps be more careful in these specificities, for example K63 chains are clearly involved in DNA repair and evidence for a role of K6 chains in mitophagy is relatively weak. Line 113-115 - something wrong here.

→ We thank the reviewer for the specific comment. We have edited the sentences.

For the non specialist S1, S1" and S2 sites should be defined.

→ We thank the reviewer for the specific comment. We have now included more detailed explanation for the Ubiquitin binding sites in the introduction (Line 116-121).

April 11, 2023

RE: Life Science Alliance Manuscript #LSA-2022-01876-TR

Prof. Donghyuk Shin
Yonsei University
Department of Systems Biology
Yonsei-ro 50
Seoul, Seoul 03722
Korea, Republic of

Dear Dr. Shin,

Thank you for submitting your revised manuscript entitled "Structural insights into ubiquitin chain cleavage by Legionella ovarian tumor deubiquitinases". We would be happy to publish your paper in Life Science Alliance pending final revisions necessary to meet our formatting guidelines.

- please upload both your main and your supplementary figures as single files
- please rename your EV figures as supplementary figures and adjust the figure callouts in the main manuscript accordingly
- please add the Twitter handle of your host institute/organization as well as your own or/and one of the authors in our system
- please add author contributions to the main manuscript text
- please remove the panel A for Supp. Fig 4; since this is the only panel in the figure, we do not need it designated with a letter
- please add a callout for Figure 6D-G, Figure S2 A-B, and Figure S3B to your main manuscript text
- Protein Data Bank accession 8GOK should be made publicly accessible at this time

A. FINAL FILES:

B. MANUSCRIPT ORGANIZATION AND FORMATTING:

Sincerely,

Reviewer #1 (Comments to the Authors (Required)):

The authors have now addressed all of the issues that I have initially pointed out and responded to my concerns. They added new experimental data, improved some of the Western blot images and included quantifications and made corrections (in text, figures and tables) where necessary. I fully recommend the publication of this nice work.

Reviewer #2 (Comments to the Authors (Required)):

The revision and response letter have adequately addressed my original, generally minor concerns, which were mostly about adding quantification and statistical analysis in certain experiments.

Reviewer #3 (Comments to the Authors (Required)):

The authors have made a sensible response to referee comments and the manuscript is significantly improved. I am happy for this to proceed to publication.

April 17, 2023

RE: Life Science Alliance Manuscript #LSA-2022-01876-TRR

Prof. Donghyuk Shin
Yonsei University
Department of Systems Biology
Yonsei-ro 50
Seoul, Seoul 03722

Dear Dr. Shin,

Thank you for submitting your Research Article entitled "Structural insights into ubiquitin chain cleavage by Legionella ovarian tumor deubiquitinases". It is a pleasure to let you know that your manuscript is now accepted for publication in Life Science Alliance. Congratulations on this interesting work.

DISTRIBUTION OF MATERIALS:

Again, congratulations on a very nice paper. I hope you found the review process to be constructive and are pleased with how the manuscript was handled editorially. We look forward to future exciting submissions from your lab.

Sincerely,
